# ZnO Nanoparticles from Different Precursors and Their Photocatalytic Potential for Biomedical Use

**DOI:** 10.3390/nano13010122

**Published:** 2022-12-26

**Authors:** Maria-Anna Gatou, Nefeli Lagopati, Ioanna-Aglaia Vagena, Maria Gazouli, Evangelia A. Pavlatou

**Affiliations:** 1Laboratory of General Chemistry, School of Chemical Engineering, National Technical University of Athens, Zografou Campus, 15772 Athens, Greece; 2Laboratory of Biology, Department of Basic Medical Sciences, Medical School, National and Kapodistrian University of Athens, 11527 Athens, Greece; 3School of Science and Technology, Hellenic Open University, 26335 Patra, Greece

**Keywords:** photodegradation, nanomaterials, zinc oxide, photocatalytic response, dye degradation, cytotoxicity, precursors

## Abstract

Semiconductor photocatalysts, particularly ZnO nanoparticles, were synthesized via the precipitation method using four different precursors (zinc acetate/zinc nitrate/zinc sulfate/zinc chloride) and compared, according to their optical, structural, photocatalytic, and anticancer properties. The materials were characterized via X-ray Diffraction method (XRD), micro-Raman, Fourier Transform Infrared Spectroscopy (FT-IR), Brunauer–Emmett–Teller (BET), Dynamic Light Scattering (DLS), and Field Emission Scanning Electron Microscope (FESEM) analysis. Photocatalysis was conducted under UV and visible light irradiation, using Rhodamine B as the organic pollutant. It was observed that the highest photocatalysis efficiency was obtained by the nanoparticles synthesized from the zinc acetate used as precursor material. A cell-dependent anticancer efficiency of the tested ZnO nanoparticles was also observed, that was also attributed to the different precursors and the synthesis method, revealing that the nanoparticles that were synthesized from zinc acetate were more bioactive among the four tested precursors. Overall, the data revealed that both the enhanced photocatalytic and biological activity of ZnO nanoparticles derived from zinc acetate precursor could be attributed to the reduced crystalline size, increased surface area, as well as the observed hexagonal crystalline morphology.

## 1. Introduction

During the last few decades, rapid industrialization, progressive urbanization, increasing population, irrational utilization of non-sustainable resources, as well as irrepressible exploitation of natural resources are leading to harmful and irreversible environmental damage [1]. The wastewater and effluents from several industries, such as textile, paint, leather, plastics, printing inks, cosmetics, food, refineries, and anthropogenic activities are putting at risk natural aquatic resources such as ponds, lakes, rivers, and oceans [2]. As a result, the clean water crisis, declared to be among the most challenging, as well as imminent, global issues, could induce adverse effects in human survival [3,4]. These effluents contain a plethora of pollutant categories, such as organic (dyes, surfactants, phenols, etc.), inorganic (metal oxides, heavy metal ions and complexes, salts, etc.), pathogens, and various others. Among these types of pollutants, the organic ones are gaining constantly increasing attention due to certain characteristics, such as (a) extensive applications and subsequent discharge to water reservoirs and land areas, (b) extended persistence, (c) intense resistance, and (d) consequential impacts both on human health as well as the environment [5].

Organic dyes are complex molecules (ionic salts) [6] adding color to fabrics and materials, the majority of them possessing resistance to detergents and heat. For a vast variety of commercial products, dyes constitute a crucial part of the industrial processes. Nowadays, more than 100,000 different dyes are commercially available, characterized by an annual production greater than 7 × 10^5^ tons [7]. Dyes are mainly water-soluble, and present low biodegradability as well as difficulty in detection at relatively low concentrations [8]. Discharging of organic dyes in water reservoirs leads to the development of unenviable color, which limits the penetration of sunlight that facilitates the fundamental activities of marine life, such as photochemical and biological activities [9]. The majority of dyes are harmful and toxic, as well as characterized as mutagenic, carcinogenic, or teratogenic, according to the FDA (Food and Drug Administration) [3,8]. The consumption of dye-contaminated water results in severe health problems and could potentially lead either to impairment of crucial human organs, such as the kidneys, brain, liver, and CNS (central nervous system), or skin irritation [8].

Rhodamine B (RhB) belongs to the xanthenes family of dyes, commonly utilized in the staining industry. Additionally, xanthenes can be exploited in biochemistry as biological markers, detecting the presence of various viruses [3]. Rhodamine B is a molecule presenting increased stability, being suspected of promoting the emergence of subcutaneous tissue sarcoma and eyes, skin, and respiratory tract irritation, as well as of being carcinogenic and mutagenic for living organisms [3]. RhB can also disturb the photosynthetic process of marine flora, as it obstructs light absorption [10].

Several techniques, such as synthesis by a facile hydrothermal, followed by the photo-deposition process can be used to produce metal oxides with ideal electrochemical properties to be exploited for energy storage, protecting the environment [11]. Various methods, such as membrane filtration, catalytic reduction, reverse osmosis, coagulation, ion exchange, oxidation, and biological methods have been also assessed towards degradation of such pollutants [12]. Advanced oxidation processes (AOPs) have been identified as an alternative technology that could be used for removing organic pollutants from water [12]. Amongst them, heterogeneous catalysis using semiconductors, such as titanium (TiO_2_) and zinc oxide (ZnO), has received great attention [13,14], as it presents the ability to degrade organic pollutants through the production of reactive oxygen species (ROS) [15]. These generated species have the potential to degrade organic pollutants into CO_2_ and H_2_O, due to their strong oxidizing nature, a process known as “mineralization” [16]. Thus, photocatalysts are able to oxidize and reduce organic pollutants on their surface, promoting their degradation [17].

Cancer is considered a multivariate disease, characterized by changes in cell processes [18]. Conventional treatments are still far from considered ideal, due to their side effects [19]. The improvement of photodynamic therapy PDT, using photocatalytic materials such as TiO_2_ or ZnO nanoparticles (NPs), might provide accurate targeting of the tumor tissues, diminishing the damage in healthy tissues. The nanoparticles can improve its permeability and retention (EPR effect) and increase the retention time in the malignant tissue [20].

Zinc oxide (ZnO) nanoparticles (NPs) constitute one of the most thoroughly studied semiconducting metal oxides as a photocatalyst. ZnO is a II-VI semiconductor, characterized by a broad band gap (3.37 eV), a wurtzite hexagonal phase at room temperature (RT), and a wide exciton binding energy (~60 meV) [21,22]. Given its extraordinary physical and chemical properties, such as increased chemical stability, enhanced electrochemical coupling coefficient, wide range of irradiation absorption, as well as high photo-stability, it can be easily classified as a multi-functional material [23]. Several applications of ZnO nanoparticles rely on controlling both physical and chemical properties, as size distribution, shape, surface state, structure, etc. [24].

In addition, ZnO nanoparticles possess interesting properties, such as biocompatibility, increased selectivity, high cytotoxicity, and facile synthesis, and are thus promising as anticancer agents. Being one of the crucial trace elements of the human organism and acting as a co-factor of over than 300 enzymes, zinc plays a vital role in supporting fundamental cellular processes, including DNA replication and repair, oxidative stress, progression of the cell cycle, and apoptosis [25]. As a result, it is evident that a slight change in zinc levels within cancer cells might lead to a deleterious effect.

Various synthesis methods, separated mainly between bottom-up and top-down techniques, can be applied in order to develop ZnO NPs [26]. ZnO nanoparticles can also be prepared from different precursor ZnO salts; e.g., ZnO NPs by precipitation using Zn(NO_3_)_2_ (zinc nitrate hexahydrate) and NaOH (sodium hydroxide) as precursors [27]; solvothermal synthesis, applying the thermal hydrolysis of Zn(CH_3_COO)_2_·2H_2_O in 1-hexadecanol smelt [28]; simple sol-gel synthesis using zinc acetate dehydrate (ZnC_4_H_6_O_4_·2H_2_O) [29]; nanoplate and flower-like motif composed of nanorods by employing heptahydrate zinc sulfate (ZnSO_4_·7H_2_O) and sodium hydroxide (NaOH) as starting materials of hydrothermal system, and urchin shape composed of ZnO nanoplates obtained utilizing Vitamin C as directing agent [30]. It is observed that utilizing varying precursors implies the development of materials with different structural and optical properties [31]. It is also observed that the concentration of zinc salts, pH variations, and reaction time affect the final structure and shape of the nanoparticles, resulting in particles with different sizes and morphologies [32,33]. ZnO nanostructures are synthesized in various forms, such as nanoparticles, nanorods, nanotubes, nanobelts, nanosprings, nanospirals, nanorings, and etc. [34].

In this study, we report an easy, low-cost, re-producible method for synthesizing ZnO nanoparticles via a precipitation method, starting with different zinc salts as precursors (zinc acetate, zinc sulfate, zinc nitrate, and zinc chloride). The ultimate goal is to optimize the synthesis protocol regarding the selected precursor, and to develop NPs with enhanced photocatalytic performance. Thus, the photocatalytic degradation of Rhodamine B (RhB) under UV and visible-light irradiation was examined using the as-prepared nanostructures. All precursors resulted in pure, hexagonal crystals (wurtzite structure); however, when comparing the prepared samples, it is observed that the ZnO nanoparticles deriving from zinc acetate are more sufficient for their photocatalytic activity against Rhodamine B and demand less time under visible or UV irradiation, in order to lead to its complete degradation. Furthermore, among the scopes of this study was the use of ZnO for targeting various cancer and normal cells, to compare the efficiency of each of the four developed types of ZnO based on the precursor that was used. A cell-dependent anticancer efficiency of the tested ZnO nanoparticles was also observed, which was also attributed to the different precursors and the synthesis method. Zinc oxide nanoparticles resulting from zinc acetate as a precursor were proven to be the most bioactive. To our knowledge, there are various studies that focus on the comparison of the ZnO synthesis protocols, and others that investigate solely the photocatalytic effect or cytotoxicity. However, the present study, for the first time according to the authors’ best knowledge, attempts to compare ZnO powders produced by all four different Zn precursors available, regarding their photocatalytic effect under UV and visible light irradiation in parallel with cytotoxicity and cell proliferation studies. The obtained results could potentially contribute to the correlation of the morphology and structural properties of ZnO nanoparticles with their photocatalytic performance and their consequent biological activity.

## 2. Materials and Methods

### 2.1. Materials and Reagents

Lab distilled water was used. Zinc acetate dihydrate (CH_3_COO_2_)Zn·2H_2_O (98%), zinc nitrate hexahydrate (Zn(NO_3_)_2_·6H_2_O) (98.5%), zinc chloride (ZnCl_2_) (98%), and zinc sulfate (ZnSO_4_·7H_2_O) (99.5%) were used as the source of Zn^2+^ cations. Zinc acetate dihydrate and zinc nitrate hexahydrate were purchased from PENTA-Manufacture of Pure and Pharmaceutical Chemicals (Penta-Chemicals Unlimited, Prague, Czech Republic). Zinc chloride was purchased from Fischer Scientific U.K. Limited (Fischer Scientific, Leicester, UK). Zinc sulfate heptahydrate was purchased from Thermofischer (Thermo Fisher (Kandel) GmbH, Kandel, Germany). Sodium hydroxide (NaOH) (98.44%) and Potassium Hydroxide (KOH) (99%), which were added to the precursor materials, were purchased from Panreac Quimica SA (Panreac Quimica, Barcelona, Spain). Ethanol of 99.8% purity was supplied by Fischer Scientific U.K. Limited (Fischer Scientific, Leicester, UK). The pollutants that were used in the experimental photocatalysis procedure is Rhodamine B (C_28_H_31_CIN_2_O_3_) (Mr = 47,902), purchased from PENTA-Manufacture of Pure and Pharmaceutical Chemicals (Penta-Chemicals Unlimited, Prague, Czech Republic).

### 2.2. Synthesis Procedures of ZnO Nanoparticles

The main method applied, in order to derive ZnO nanoparticles, was the precipitation method. Different procedures for each precursor, though very similar to each other, were conducted. The main difference was the solution added in each precursor, which varied between sodium hydrate (NaOH) and potassium hydrate (KOH), and their concentration range. Figure 1 represents the main steps of the synthesis procedure.

#### 2.2.1. Synthesis of ZnO Nanoparticles, Using Zinc Acetate Dehydrate Precursor

The first precursor that was used to prepare ZnO nanoparticles was zinc acetate dihydrate (CH_3_COO_2_)Zn·2H_2_O (M_r_ = 219.5 g/mol), according to the method reported by Madhavi and Ashraf Talesh [35]. Firstly, a 0.15 M zinc acetate solution was prepared by adding zinc acetate dihydrate to distilled water, and then it was magnetically stirred until it became uniform. The pH value measured at the start of the synthesis was 6.65. According to the procedure, sodium hydroxide (NaOH) was added dropwise to the point where the solution became white. The pH value at that point was 13. After 2 h at 50 °C, the solution was centrifuged three times for 10 min at 9000 rpm, in order to recover the white precipitate. In order to remove any organic remains, the precipitate was centrifuged with water two times and finally with pure ethanol. The precipitate was left at 100 °C for an hour to dry, and then annealing was realized at 400 °C for 3 h. The final product, for convenience purposes, will be called hereinafter ZnAc.

The synthesis reaction that took place was given by chemical reaction (1) [36]
Zn(CH_3_COO)_2_ + 2NaOH → ZnO + 2CH_3_COONa + H_2_O(1)

#### 2.2.2. Synthesis of ZnO Nanoparticles, Using Zinc Nitrate Hexahydrate Precursor

The steps to manufacture ZnO NPs from zinc nitrate hexahydrate (Zn(NO_3_)_2_·6H_2_O) (M_r_ = 297.49 g/mol) as precursor were similar to the previous, according to Suntako, 2015 [27]. Instead of NaOH, in this case, KOH was added drop by drop. The concentrations of the precursors must be the same, so as to compare them accurately. A 0.15 M zinc nitrate solution was made by adding zinc nitrate hexahydrate into distilled water. The procedure was the same as above, but instead of 400 °C, the recovery was annealed for 3 h at 500 °C. It is noteworthy that the quantity of the recovery in this synthesis was higher than the others. The final product, for convenience purposes, will be called hereinafter ZnNit.

The synthesis was realized through the chemical reaction (2) [36]
Zn(NO_3_)_2_ + 2KOH → ZnO + 2KNO_3_ + H_2_O(2)

#### 2.2.3. Synthesis of ZnO Nanoparticles, Using Zinc Chloride Precursor

ZnO nanoparticles were prepared, using 0.15 M zinc chloride (ZnCl_2_) (M_r_ = 136.29 g/mol) as precursor, according to Bacaksiz et al., 2008 [31]. At first, zinc chloride was added in distilled water and pH was adjusted with the addition of NaOH. The solution was left stirring at room temperature for 2 h. The centrifuge process followed and was repeated three times; once by washing the sediment with ethanol and twice with distilled water. The precipitate was left at 100 °C for an hour to dry and then annealing was performed at 400 °C for 3 h. The amount of the nanoparticles recovered was much less than the aforementioned methods of synthesis. The final product, for convenience purposes, will be called hereinafter ZnChlor.

Chemical reaction (3) [36] describes the synthetic procedure of this type of ZnO.
ZnCl_2_ + 2NaOH → ZnO + 2NaCl + H_2_O(3)

#### 2.2.4. Synthesis of ZnO Nanoparticles, Using Zinc Sulphate Heptahydrate Precursor

Zinc sulfate heptahydrate (M_r_ = 287.6 g/mol) was used as a precursor for a ZnO synthesis too, according to Limón-Rocha et al., 2019 [37]. A solution of 0.15 M was prepared by mixing zinc sulfate heptahydrate with distilled water. After the solution became uniform, NaOH 1 M was added drop by drop, until the solution obtained a milky-white color, and then it was left stirring for 24 h. The white precipitates acquired were centrifuged three times, once with ethanol 99.8% and twice with distilled water. The white powder was dried for 1 h at 70 °C, and then annealed for 4 h at 400 °C, following the instructions of Kahouli et al., 2018 [38]. The final product, for convenience purposes, will be called hereinafter ZnSulf, and it was prepared through chemical reaction (4) [36]:ZnSO_4_ + 2NaOH → ZnO + Na_2_SO_4_ + H_2_O(4)

After the annealing, the white material received through each of the previously mentioned synthesis methods were stored in brown-colored glass jars, so as for it not to be unintentionally photoactivated.

### 2.3. Rhodamine B Solution

Rhodamine B (RhB) is the pollutant that was used in the present study to compare the photocatalytic activity of the different precursor samples. Purified Rhb (Penta chemicals (Czech Republic) were added to distilled water to prepare the Rhb solution.

### 2.4. Characterization Methods

The X-ray Diffractometer, used to study the crystallinity of the nanoparticles, was a Brücker D8 Advance (D8 Advance, Bruker, Germany) with a CuΚα radiation (30 kV, 15 mA). The measurement was conducted at 2 theta angle between 20 and 80° and counting the diffraction intensity every 0.01° per 0.5 s. A spectroscopic technique that is often used to characterize wide band gap semiconductors, such as ZnO nanoparticles, is Raman, which can be used to examine the structural quality of the samples [24]. Specifically, defects, crystallinity, and impurities can be detected. ZnO exhibits a wurtzite structure that crystallizes in the hexagonal space group C^4^_6v_ (P6_3_mc). The primitive cell includes two formula units where all atoms are occupying the 2b sites of the C3_v_ symmetry. The vibration modes of normal network to the Γ point of the Brillouin zone C can be generally predicted by the equation of optic group theory Γ = A_1_ + 2E_2_ + E_1_ + 2B_1_, where A_1_ and E_1_ constitute polar modes attributed to motions of atoms, parallel (A_1_) and perpendicular (E_1_) to the c-axis, respectively, while both of them split into one transverse optical (TO) and one longitudinal (LO) mode. The two non-polar modes E_2(Low)_ and E_2(High)_ are ascribed to the vibration of Zn and Oxygen sub-lattice, respectively. B_1(silent)_ modes are Raman-inactive, while E_2_, A_1(TO)_, and E_1(TO)_ modes are observed during an incident radiation perpendicular to the sample’s c-axis. On the other hand, E_2_ and A_1(LO)_ modes are detected during an incident light parallel to c-axis [31]. Raman spectra were conducted at room temperature. The micro-Raman apparatus that was used in the present study was functioned at a wavelength of 532 nm from a solid-state laser (λ = 532 nm), with average power of around 50 mW (inVia, Renishaw, Wotton-under-Edge, Gloucester, UK). The frequency shifts were calibrated by an internal Si reference. A number of 2 to 10 spots were acquired for each sample. The exposure time was 30 s, with 3 accumulations, 0.1% power, and a range of 100–1500 cm^−1^.

FT-IR defines the composition and quality of the product and can be potentially used for the absorbed ZnO nanoparticles. The FT-IR instrument used was the FTIR JASCO4200 (Interlab, Athens, Grrece), using Ge crystal. FT-IR spectra of ZnO nanopowder was recorded between 400 and 4000 cm^−1^. N_2_ adsorption was measured in a ChemBET 3000 Instrument (Yumpu, Diepoldsau, Switzerland) to determine the BET specific surface area. Firstly, each sample underwent a degassing procedure at 80 °C for 24 h.

Dynamic Light Scattering (DLS) was applied to measure the size (hydrodynamic diameter) and distribution of the particles suspended in water solutions. The incident light was a 633 nm laser and a scattering angle of 173° was used for recording scattering intensity (Malvern Zetasizer Nano ZS, Malvern Panalytical Ltd., Malvern, UK). Energy band gap (*E_g_*) was evaluated via an ultraviolet–visible (UV–Vis) spectrometer (Jasco UV/Vis/NIR Model name V-770, Interlab, Athens, Grrece) that was equipped with an integrating sphere, allowing diffuse reflectance measurements. Characterization of the morphology (nanorods, flowers, etc.) was performed using Field Emission SEM (FESEM, JSM-7401F, JEOL, Tokyo, Japan).

### 2.5. Photocatalytic Degradation of Rhodamine B

The evaluation for photoactivity was tested using 0.075 g of the catalyst (ZnO NPs) in 10 mg·L^−1^ Rhodamine B solution (200 mL) at room temperature. The pollutant solution was prepared by stirring 0.0017 g of Rhodamine B with 200 mL of distilled water for 2 h. Before the experiment, O_2_ was passed through the Rhodamine B solution for 60 min to saturate it. In order to fully examine the photocatalytic potential of the synthesized ZnO nanoparticles, the experiments were carried out under irradiation of both visible and UV light. The photoreactor used has four parallel lamps, at a distance of 10 cm over the samples [39,40]. The lamps used were 15 W visible lamps of 900 lumens (made by Osram, OSRAM GmbH, Munich, Germany) and blacklight of 368 nm lamps of 830 lumens, made by Sylvania (Sylvania, Wilmington, NC, USA). All the tests were conducted at room temperature and pH equal to 6.4. The degradation of Rhodamine B was inspected, under each light source used. The absorbance of the samples was measured using the Hitachi U-2001 Spectrophotometer (Hitachi, Tokyo, Japan). The evaluation of the ratio of the measured A (absorption at each time) to the initial (A_initial_) allowed the determination of the ratio *C*/*C_o_* [39]. After a few hours upon irradiation, the degradation can be observed by watching the color progressively turning lighter, until it turns completely transparent.

#### Total Organic Carbon (TOC) Analysis

The mineralization of RhB was evaluated, employing the total organic carbon (TOC) analysis for the same time points as used during the photocatalytic test.

### 2.6. Photo-Induced Biological Effect

#### 2.6.1. Cell Cultures

A549 human epithelial lung carcinoma (LGC Standards GmbH, ATCC, Wesel, Germany, CCL-185™), MDA-MB-231 (ATCC, HTB26™), human epithelial breast adenocarcinoma (metastatic), and human foreskin derma fibroblast cell strains FF95 were cultured in Dulbecco’s modified Eagle’s medium (DMEM) (Gibco BRL, Life Technologies, ThermoScientific, Paisley, UK). The medium was supplemented with 10% fetal bovine serum (FBS), 1% L-glutamine, 1% sodium pyruvate, and antibiotics (Gibco BRL, Life Technologies, Thermo Scientific, Paisley, UK). The cultures were kept at 37 °C, 95% humidity, in a 5% CO_2_ incubator. When trypisinized, cells were detached with trypsin/EDTA for 3 min at 37 °C before resuspending the cells in the medium [41,42].

#### 2.6.2. Effect on Cell Proliferation

Cells (~100,000 cells/well) were treated with increasing concentrations of the four ZnO powders 24 h after plating and irradiated with visible light for the first series of experiments, and with UV light for the second one. Staining with Trypan Blue and counting using a hemocytometer (Neubauer, Corning, The Netherlands) through an optical microscope (OLYMPUS IM, Olympus Deutschland GmbH, Hamburg, Germany) were conducted every day, in order to prepare graphs depicting the growth rates. The experiment was repeated three times in triplicates. Kruskal–Wallis non-parametric test was applied and *p* < 0.05 was considered statistically significant [39].

#### 2.6.3. Cytotoxicity Test

In order to estimate the cell viability, MTT colorimetric assay (3-(4,5-dimethylthiazol-2-yl)-2,5-diphenyl-tetrazolium bromide) (Thiazolyl Blue Tetrazolium Bromide M5655, Sigma-Aldrich, Darmstadt, Germany) was performed, as reported previously [43]. For the control group, the cells used were treated with 0.5 mg/mL of cis-platin. The process was repeated three times [42,43]. Kruskal–Wallis non-parametric test Statistical analysis was applied. *p* < 0.05 was considered statistically significant.

## 3. Results

### 3.1. Characterization of the Nanoparticles

#### 3.1.1. XRD Analysis

The indexed peaks in the obtained spectra for all the synthesized ZnO particles, through utilization of different zinc precursors, are completely identified with that of bulk ZnO ((JCPDS) Card No. 36–1451) [44], confirming monocrystallinity, as well as a wurtzite hexagonal structure for all prepared samples [45] (Figure 2). Νo other peak related to impurities was detected in the spectra within the detection limit of the XRD, certifying that the synthesized powders are pure ZnO [46,47]. The fact that there are strong and narrow diffraction peaks indicates that all the ZnO nanoparticles had a well-crystalline structure, according to Rafaja et al. [48].

The formed peaks at 2*θ*: 31.74°, 34.42°, 36.26°, 47.57°, 56.63°, 62.89°, 66.43°, 67.95°, 69.07°, 72.69°, and 76.86° correspond to the (100), (002), (101), (102), (110), (103), (200), (112), (201), (202), and (004) (Miller indices) planes, respectively. Particularly, crystal lattice indices and crystallinity determination, as well as average crystallite size and interplanar d-spacing calculations, were conducted. The results are presented in detail in Table 1, Table 2, Table 3, Table 4 and Table 5.

The average crystallite size of the ZnO powders was evaluated by the Debye–Scherrer equation (Equation (1)):
(1)$$D=\frac{0.89\lambda }{\beta cos\theta }$$
where *λ* = X-ray wavelength (*λ* = 1.5406 A°), 0.89 is Scherrer’s constant, *β* stands for Full Width at Half Maximum (FWHM) of the peak that was related to (101) plane, and *θ* is Bragg’s angle [37].

Furthermore, interplanar d-spacing was calculated through Bragg’s Law Equation (Equation (2)):(2)2dsinθ=nλ,n=1

Crystallinity index (*CI*%) was estimated according to Equation (3):(3)$$CI\;\left(\%\right)=\frac{Area\;of\;all\;the\;crystalline\;peaks}{Area\;of\;all\;the\;crystalline\;and\;amorphous\;peaks}$$

Crystal lattice index was determined using Equation (4) [49]:(4)$$\frac{1}{d_{hkl}^{2}}=\frac{h^{2}}{a^{2}}+\frac{k^{2}}{b^{2}}+\frac{l^{2}}{c^{2}}$$

There are no significant changes in the peak positions, as is obvious in the XRD ZnO diffractograms. However, diversification among the intensity of the different samples can be observed, which can affect the FWHM, resulting in a different crystallite size.

The (101) plane exhibited the highest relative intensity for the entire XRD pattern, suggesting anisotropic growth and preferred orientation of the crystallites, since epitaxial growth along the C-axis of the (001) direction is a typical phenomenon for wurtzite structured materials [50].

According to our results, among the produced samples, ZnAc possesses the smallest average crystallite size (4.27 nm), also showing the highest crystallinity.

#### 3.1.2. Micro-Raman Analysis

The Raman spectra obtained from the produced ZnO powders are presented in Figure 3. Similar spectra were acquired through the different synthetic procedures. The Raman features in ZnO powder are attributed to Raman active modes of the ZnO wurtzite crystal [51].

The highest Raman intensity peak in the spectrum of ZnO powder is that at ∼440 cm^−1^ (E_2H_ mode), ascribed to oxygen vibration [51]. Its pronounced asymmetry is associated with the lattice disorder, as well as to the anharmonic phonon–phonon (P–P) interactions [45]. Opposite to E_2_ phonons, polar phonons A_1_ and E_1_ are both split into TO and LO phonons. Thus, this sharp peak is a characteristic Raman active peak for the wurtzite hexagonal phase of ZnO [38]. It should be mentioned that the E_1(L)_ mode at ∼590 cm^−1^ presents background rise, attributed to second-order Raman scattering. The existence of impurities and/or defects can strongly influence the E_1(L)_ mode [51], particularly for ZnSulf and ZnChlor.

The second-order mode observed in the low-wavenumber region, at ∼335 cm^−1^, is ascribed to difference E_2H_−E_2L_ [45]. In the intermediate region of the spectrum of non-activated ZnO powder, the second-order Raman modes with A_1_ symmetry (484, 701, 725, and 742 cm^−1^) are weak and hardly noticed, except for the mode at 541 cm^−1^. In addition, combination of acoustical and optical modes occurs at ∼660 cm^−1^.

#### 3.1.3. FT-IR Analysis

On the FT-IR spectra (Figure 4), the ZnO peak is detected at 554 cm^−1^ for the sample made by zinc acetate precursor, 544 cm^−1^ for the sample made from the zinc nitrate precursor, 546 cm^−1^ for the sample made from the zinc chloride precursor, and 540 cm^−1^ for the sample made from the zinc sulfate precursor. Inter-hydrogen bond is associated with the centered peak of 3478 cm^−1^, which indicates the existence of water molecules and hydroxyl group. This is probably due to the atmospheric humidity that affects the samples when exposed for measurement. Our findings are in agreement with several other studies focusing on the investigation of the properties of ZnO [52,53].

The peak at 1113 cm^−1^ is attributed to the C-O stretching of primary alcohols. Two more sharp peaks are observed, and are most notably present on the FT-IR spectra of the sample constructed by the zinc acetate precursor. The peak at 1520 cm^−1^ corresponds to a vibration of C=O bond. This can be caused by a presence of organic residues, which might have remained even after the reaction with the acetate. The peak at 1427 cm^−1^ might correspond to the carboxylic acid O-H bending. The detected peak at 2351 cm^−1^ is assigned to carbon dioxide O=C=O stretching. Our findings are in accordance with other studies focusing on ZnO [52].

#### 3.1.4. BET Specific Surface Area Analysis

The obtained plots from the BET method are given on Figure 5. Some of the N2-sorption plots show a crossover of the sorption (black line) and desorption (red line). That crossover means that the material is being affected by the measurement [54]. For those materials, BET surface area values are reliable, because they are calculated from the early part of the adsorption. It is observed that the zinc sulfate gives less reliable results; however, it is in an acceptable range.

Results of the BET Method regarding surface area, micropore surface area, cumulative volume and average pore diameter are presented in Table 6.

ZnAc exhibits the highest surface area value, implying that it allows more active sorption sites to occur during photocatalysis experiments in comparison with the other three types of ZnO powders. This fact is perhaps associated with the relatively small average crystallite size of ZnAc, compared to the other powders [50].

#### 3.1.5. DLS Analysis

The DLS measurements were realized at a set value of pH value ~7. The hydrodynamic radius distribution as a function of scattering intensity is shown in Figure 6 for all synthesized ΖnO particles. The results show that the particle samples prepared by zinc acetate show a better distribution. The size of all the produced NPs was in the range of 10–100 nm. ZnNit, ZnChlor, and ZnSulf show a good distribution; however, small particles in the size of microns μ are also detected, probably due to the existence of agglomerates (Figure 5). The coefficient of variation (CV) was ~0.99%. Pourrahimi et al. had prepared high purity ZnO nanoparticles by using acetate salt with a mean size of ~25 nm [55], in agreement with our experimental findings. The obtained data from DLS are presented in Table 7, for all four samples.

#### 3.1.6. Diffuse Reflectance UV–Vis Spectroscopy

Band gap energy (*E_g_*) is a critical parameter and should be taken into account in photocatalytic studies. All the tested powders showed absorbance bands, with an absorption edge below 385 nm that is characteristic for the wurtzite crystal phase of ZnO and are of high intensity. The aforementioned bands correspond to O_2p_→Zn_3d_ electron transition from the valence band (VB) to the conduction band (CB).

Kubelka–Munk (K–M) was applied to measure the reflectance of the ZnO powders, according to Equation (5) [39,56]:(5)$$F\left(R\right)=\frac{{\left(1-R\right)}^{2}}{2R}$$
where R is the reflectance. The reflectance changes in the ZnO powders within the light spectrum are presented in Figure 7a.

Figure 7b depicts the direct *E_g_* of the produced ZnO powders, applying the K–M model vs. energy by extrapolating the linear region of the spectra (F(R)*hv*)^1/2^ vs. *hv*. *E_g_* was calculated by Tauc’s equation (Equation (6)):(6)$$ahv=A{\left(hv-E_{g}\right)}^{n}$$
where *E_g_* is the energy band gap, *h* is Planck constant, *v* stands for the frequency, *α* is the absorption coefficient, and *n* = ½ [39]. There is no significant difference between *E_g_* values among the tested powders, a result found in agreement with the related literature, namely *Eg* = ~3.37 eV [22].

#### 3.1.7. FESEM Analysis

The main morphological characteristics of the produced powders observed by using Field Emission Scanning Electron Microscope (FESEM) are shown in Figure 8.

The nanoparticles derived from zinc chloride, zinc nitrate, and zinc sulfate are flake-shaped, while those derived from zinc acetate exhibit a combination of spherical and hexagonal particles. It is well known that ZnO is found in various morphologies, including 0D, 1D, 2D, and 3D structures. The most common ones are hexagonal and diamond-shaped; thus, our findings are in accordance with the related literature [57]. Since the synthesis procedure was conducted under the same conditions, including thermal treatment, the observed morphological differences among the tested powders might be attributed to the different precursors utilized.

All the produced samples seemed to be nanostructured, as can be perceived through the FESEM acquired images, and also can be considered as adequately homogenous.

### 3.2. Photocatalysis of Rhodamine B

#### 3.2.1. Photocatalytic Activity

The photocatalytic activity of the produced ZnO (ZnAc, ZnChlor, ZnNit, ZnSulf) powders was evaluated through the degradation of Rhodamine B (RhB) under visible and UV irradiation. Additionally, real time UV–Visible spectra were acquired for both visible and UV light photocatalytic treatments and presented in Figure 9 and Figure 10, respectively. Both photolysis (RhB photolysis) and adsorption–desorption equilibrium (RhB dark), for the same duration as the procedure, under no irradiation (dark condition), as well as under continuous stirring, were additionally conducted. Utilizing either visible or UV irradiation (Figure 11 and Figure 12), the amount of Rhodamine B degraded was less than 1% during the photocatalytic procedure, indicating that the degradation percentage of RhB in the absence of each examined photocatalyst is extremely low. Same results derived from the test under dark conditions, confirming the stability of RhB dye.

During the tests, the ZnAc photocatalyst presented the highest efficiency among all examined ZnO powders, reaching total degradation of RhB at 210 min under visible light, whereas it needed less than 100 min to decompose Rhodamine B utilizing UV irradiation. The visible light photocatalytic activity of bare ZnO could be attributed to the oxygen vacancies in the lattice of ZnO, as they allow more photons to be trapped [58]. According to various studies, ZnO can sufficiently, photocatalytically degrade organic pollutants and dyes such as methyl orange [59], methylene blue [60], and RhB [61] under visible light irradiation.

Additionally, it was observed that, in contrast with visible light, which required at least 4.5 h, the complete photodegradation under UV light irradiation, by adding ZnSulf to the pollutant, needed only 3 h to be achieved. ZnChlor required more than 4 h to completely degrade Rhodamine B under UV light, whereas ZnSulf required only 2.5 h to do so.

The high photocatalytic performance of ZnAc compared to the other powders can be attributed to the relatively smaller average crystallite size, as it was obtained through XRD analysis; the higher surface area, as was shown via BET measurements; the smaller hydrodynamic diameter (D_h_), as was indicated through DLS; and the hexagonal morphology, as was shown by FESEM micrographs. It is known that as the surface area increases, light harvesting capacity and faster interfacial charge transfer rates also increase significantly, while a higher contact area among tested catalysts and dye molecules could be established, causing a maximum adsorbent effect. This could be useful in promoting a large number of active sites for the effective interaction among a ZnAc catalyst and RhB molecules [62]. These characteristics are the main points of difference compared to the other ZnO powders.

#### 3.2.2. Photocatalytic Kinetic Model Study

Figure 13 illustrates the results of the kinetic model studies under visible light. The rate of the photocatalytic absorption of RhB onto the nanopowders’ surface is relatively faster for ZnAc, based on the pseudo-first order kinetics (0.017 min^−1^), than the rest of the samples. It is worth mentioning that, taking into account the R^2^ values of the kinetic studies, the adsorption process fits well to the pseudo-first order kinetics (Figure 13a) compared to pseudo-second order kinetics (Figure 13b), where R^2^ values are presented rather poorly. Table 8 presents the kinetic parameters of the samples.

The pseudo-second order equation is given below (Equation (7)):(7)$$\frac{t}{q_{t}}=\frac{1}{k_{2}q_{e}^{2}}+\frac{1}{q_{e}}t$$
where *q_t_* and *q_e_* stand for the pollutant amount adsorbed at time t and equilibrium, respectively (mg/g), and *k*_2_ is the rate constant (g·mg^−1^·min^−1^).

Similarly, a pseudo-first order kinetic model is accepted for the case of photocatalytic tests under UVA irradiation (Figure 14 and Table 9).

#### 3.2.3. Mineralization of RhB

The percentage of mineralization of the RhB was evaluated using Equation (8):(8)$$Mineralization\;=1-\frac{TOC_{final}}{TOC_{initial}}\cdot 100$$
where *TOC_initial_* is the total organic carbon concentration in the medium before the photocatalysis process and *TOC_final_* is the total organic carbon concentration in the medium after the photocatalysis reaction [50]. The obtained data are presented for the visible and UV light photocatalysis treatment in Figure 15a,b, respectively. According to the results obtained through the TOC analysis, ZnAc powder presents the highest percentage of RhB mineralization under both visible and UV light irradiation. These findings are in congruence with the results of RhB degradation studies (Figure 10 and Figure 12).

#### 3.2.4. Photocatalysis Mechanism

The main steps of the heterogeneous photocatalytic degradation of RhB by ZnO (Figure 16) can be described as follows: (a) RhB diffuses from the liquid phase and sticks to the surface of the photocatalyst. (b) RhB is adsorbed on the ZnO surface. (c) Chemical reactions (oxidation and reduction reactions) take place in the adsorbed phase (Figure 17). The products of this process are desorbed. (d) The products are removed from the photocatalyst interface [63].

More analytically, when ZnO is photo-activated with photonic energy equal or greater than the energy band gap (*E_g_*), electrons (e^−^) from the VB are transferred to the CB. Thus, pairs of electrons and holes (e^−^/h^+^ ) are created and moved to the surface of the photocatalyst [64] (Figure 16). There, during the short time period that are separated, they can be involved in the redox reactions that are previously described, they can be involved in the redox reactions that are previously described (Figure 18).

### 3.3. Photo-Induced Biological Effect

#### 3.3.1. Effect on Cell Proliferation

A549 (human epithelial lung carcinoma), MDA-MB-231 (human epithelial breast adenocarcinoma, metastatic), and human foreskin derma fibroblast cell strains FF95 were cultured in increasing concentrations (0, 0.05, 0.1, 0.15, 0.2 mg/mL) of the four ZnO powders, and also irradiated either with visible light or with UV light, in order to also compare the photocatalytic efficiency of the samples. Cell counting every day until 72 h monitored the growth rates of these cell populations.

We observed that all of the tested powders, even after photo-activation, did not affect the growth rate of FF95 (Figure 19). ZnNit, ZnChlor, and ZnSulf did not affect the cell proliferation of MDA-MB-231 cells, while photo-activated with UV light ZnAc decreased the cell population by 25% (Figure 20). Only a slight, not significant decrease was detected in cell number of A549 cells upon ZnNit addition (0.2 mg/mL), while ZnChlor and ZnSulf did not change the cell number. ZnAc significantly decreased the cell population of A549 (30%) at the concentration of 0.2 mg/mL. After the photocatalytic procedure with visible light, a further decrease was presented for 0.2 mg/mL and earlier at 48 h, while 0.15 mg/mL also had an important effect, detected at 72 h. Photo-activation with UV light further amplified the same effect, decreasing the cell number to 200,000 cells at the final measurement, indicating a 50% decrease compared to untreated cells (Figure 21).

#### 3.3.2. Effect on Cytotoxicity

In order to compare the cytotoxicity of the four different types of ZnO NPs, cells treated with increasing concentrations of those samples (0, 0.05, 0.1, 0.15, 0.2 mg/mL) and irradiated when needed either with visible light or with UV light, in order to also compare the photocatalytic efficiency of each of the compounds under these two different irradiation conditions. The ultimate goal is to optimize our synthesis method regarding the selected precursor, and to develop NPs with the better photocatalytic performance.

Through our repeated experiments, we observed that ZnNit, ZnChlor, and ZnSulf did not affect the cell viability of any of the three tested cell lines, FF95, MDA-MB-231, and A549 (Figure 22b–d). However, the addition of 0.2 mg/mL ZnAc decreased the percentage of cell viability to 81% in A549 cells and decreased the cell viability of MDA-MB-231 cells by 10% (Figure 22a). The decrease in A549 was considered statistically significant, according to the Kruskal–Wallis Test. He et al. also demonstrated that ZnO NPs could induce cytotoxicity to A549 cells, which was related to increased intracellular Zn ions [65]. Additionally, Aljabali et al. indicated that ZnO NPs had a significant cytotoxic effect, associated with apoptosis on MDA-MB-231 cells [47]. Furthermore, Kc et al. showed that cell viability assay showed concentration dependent cytotoxicity of ZnO NPs in breast cancer cell line (MDA-MB-231) [66].

To evaluate the effect of the photo-activated ZnO NPs two series of experiments were realized. In our first approach, the cells were irradiated with visible light for 40 min, in the presence of increasing concentrations of the four samples (0, 0.05, 0.1, 0.15, 0.2 mg/mL). During the second series of experiments, the cells were irradiated with UV light for 10 min, in the presence of the same concentrations of the four types of ZnO NPs. These experiments would allow one to understand if the synthesis method might have any effect on the photocatalytic behavior of the produced materials, and to compare the photocatalytic performance of ZnO NPs, irradiated by different light sources. Due the wide use of ZnO in cosmetology and particularly in sunscreens, it is reasonable to investigate the cytotoxic effect of it under irradiation [67].

Photo-activated ZnNit, ZnChlor, and ZnSulf with visible light did not affect the cell viability of any of the treated cells FF95, MDA-MB-231, or A549 (Figure 23b–d). On the contrary, the photo-excitement of ZnAc led to a further decrease in MDA-MB-231 by 10% compared to the effect of non-irradiated ZnAc at the concentration of 0.2 mg/mL (Figure 23a). Additionally, there was an enhanced effect of photo-activated ZnAc on A549, since the total decrease is approximately 30% in the presence of irradiated 0.2 mg/mL (Figure 23a). Additionally, a minor decrease of ~15% was observed also in A549 cells, post addition of 0.15 mg/mL, upon irradiation with visible light.

Various studies have focused on the cytotoxic effect of ZnO. Xia et al. had attributed the cytotoxicity of ZnO to particle dissolution, and consequently to Zn^2+^ release [68]. Others explained the same effect through other indirect effects, such as labile zinc complexes, metal composition, physicochemical properties, and particularly particle size [69].

Photo-activated ZnChlor and ZnSulf with ultraviolet light did not decrease the cell viability of any of the treated cells FF95, MDA-MB-231, or A549 (Figure 24c,d). ZnNit slightly decreased the cell viability of A549 cells at the concentration of 0.2 mg/mL (Figure 24b) and after the photo-activation with UV, without affecting the other cell lines. Photo-excited ZnAc NPs succeeded in decreasing the cell viability of MDA-MB-231 and A549 by 30% and 42%, respectively, at a concentration of 0.2 mg/mL (Figure 24a). Additionally, a statistically significant reduction in cell viability was observed for A549 cells, at 0.1 and 0.15 mg/mL of photo-activated ZnAc. Additionally, Lestari et al. observed also a cytotoxic effect on MCF-7 breast cancer cells upon irradiation with UV light [70]. Additionally, Yang et al. demonstrated that both visible light and UV irradiation enhanced the cytotoxic effect of ZnO NPs on the A549 cell line, with this effect being more intense in the case of UV irradiation [71].

## 4. Conclusions

Within the framework of this study, we reported the synthesis of ZnO nanoparticles through a precipitation method, utilizing four different Zn precursors (zinc acetate dihydrate, zinc nitrate hexahydrate, zinc chloride, and zinc sulfate heptahydrate) that were compared regarding their morphology, structural and optical properties, as well as their photocatalytic and anticancer efficiency. It was clearly observed that the use of different precursor materials had a considerable impact on the final properties and efficiency of the produced ZnO powders.

FESEM morphology revealed a combination of hexagonal and spherical particles in the case of ZnAc powder, while ZnNit, ZnChlor, and ZnSulf powders were characterized by a flake-shaped morphology. XRD diffractograms of the synthesized ZnO powders indicated sharp, crystalline, wurtzite hexagonal phases of ZnO for all the samples, presenting an average crystallite size in the range of 4.2–8.2 nm. ZnAc is characterized by the smallest crystallite size (4.27 nm), showing also the highest crystallinity (81.99%). The XRD results are also verified through Raman spectroscopic analysis. DLS measurements demonstrated that ZnAc nanoparticles had a narrower distribution regarding mean size among all the examined powders (~27 nm). BET measurement results revealed that the zinc acetate precursor sample exhibited the most increased specific surface area (11 m^2^g^−1^), while the FTIR spectra analysis also verified the vibrational modes of ZnO. Finally, Diffuse Reflectance UV–Vis Spectroscopy was conducted to estimate the energy band gap (*E_g_*) that was found to be among 3.37 eV and 3.4 eV for all the samples.

ZnO nanoparticles were studied for the photocatalytic degradation of Rhodamine B. As for their photocatalytic properties, it is noteworthy that the most efficient samples were those which were prepared from the zinc acetate dihydrate and the zinc nitrate hexahydrate precursors. When applied to the Rhodamine B solution, under visible light, the zinc acetate precursor sample needed three hours to fully degrade it, in contrast with the sample made by zinc chloride, which needed apparently more than four and a half hours to do the same. Under UV light, results were similar, but the action of the nanoparticles is much more drastic; the zinc acetate sample needed less than two hours to degrade the pollutant, and the zinc chloride required more than four hours to do so. Thorough kinetic studies and mineralization trough TOC confirmed the photocatalytic degradation of the RhB.

Additionally, among the scopes of this study was the use of ZnO as therapeutic agents targeting cancer and normal cells. Thus, the different synthesis methods based on the selected precursor were further compared regarding their efficiency on cancer cell lines A549, MDA-MB-231, and also on normal fibroblasts FF95. ZnChlor and ZnSulf had no significant effect on cell viability and the cell proliferation of the treated cells, even upon irradiation with visible light or UV. Photo-excited with UV light ZnNit had a minor effect on the viability of A549 cells, while visible light irradiation did not enhance its cytotoxic potential. ZnAc NPs decreases cell viability by 20% in A549 cells and by 10% in MDA-MB-231 cells. Phot-activated ZnAc NPs with visible light further reduced the viability of metastatic cancer cells MDA-MB-231 by 10% compared to non-activated ones. Additionally, photocatalytic activation of ZnAc NPs with UV light was proven to be more effective in decreasing cell viability, even in lower concentrations of A549, and of MDA-MB-231. Thus, a cell-dependent anticancer efficiency of the tested ZnO nanoparticles was also observed, that was also attributed to the different precursors and the synthesis method. Zinc oxide nanoparticles resulting from zinc acetate as a precursor were proven to be the most bioactive.

In summary, the obtained data lead to the assumption that the enhanced photocatalytic, as well as biological, activity of ZnO nanoparticles that have been synthesized using zinc acetate as a precursor could be attributed to the reduced crystalline size, increased surface area, and the observed hexagonal crystalline morphology.

## Figures and Tables

**Figure 1 nanomaterials-13-00122-f001:**
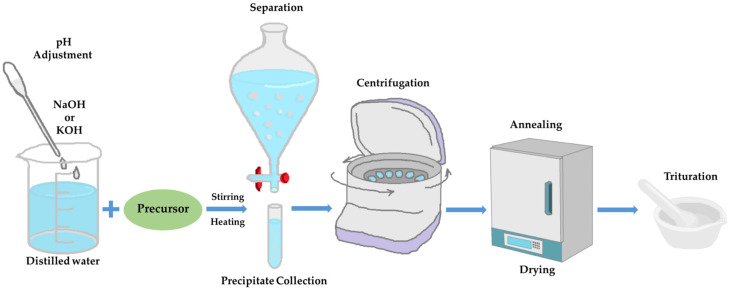
Synthesis process schematic representation. Each procedure is being separately described below.

**Figure 2 nanomaterials-13-00122-f002:**
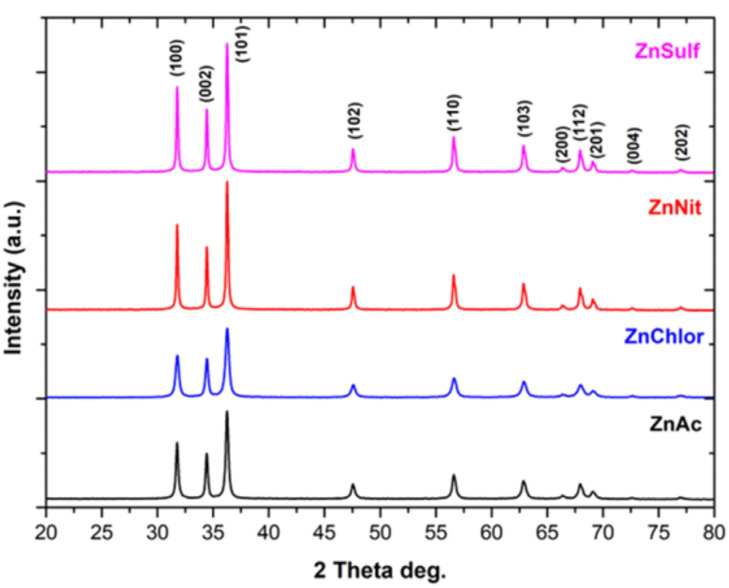
XRD diffractograms of ZnO nanoparticles, synthesized using as precursor zinc acetate dehydrate (in black) (ZnAc), zinc chloride (in blue) (ZnChlor), zinc nitrate hexahydrate (in red) (ZnNit), and zinc sulphate heptahydrate (in purple) (ZnSulf).

**Figure 3 nanomaterials-13-00122-f003:**
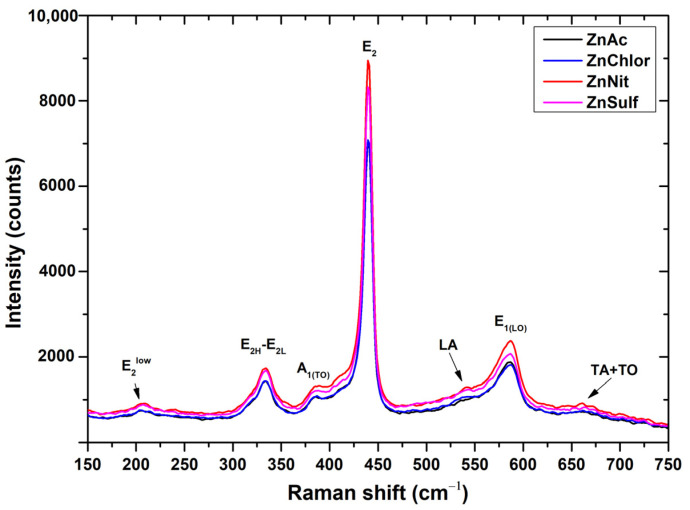
Raman spectra of ZnO nanoparticles, synthesized using as precursor zinc acetate dehydrate (in black) (ZnAc), zinc chloride (in blue) (ZnChlor), zinc nitrate hexahydrate (in red) (ZnNit), and zinc sulphate heptahydrate (in purple) (ZnSulf).

**Figure 4 nanomaterials-13-00122-f004:**
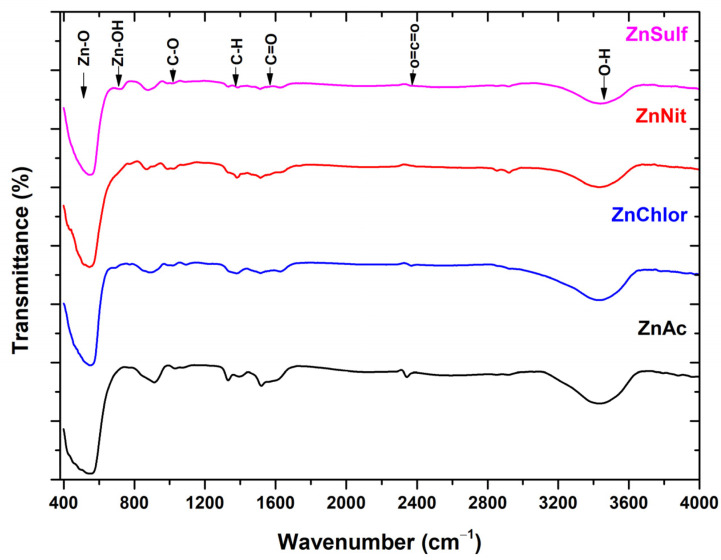
FT-IR spectra of ZnO nanoparticles, synthesized using as precursor zinc acetate dehydrate (ZnAc) (in black), zinc nitrate hexahydrate (ZnNit) (in red), zinc chloride (ZnChlor) (in blue), zinc sulphate heptahydrate (ZnSulf) (in purple).

**Figure 5 nanomaterials-13-00122-f005:**
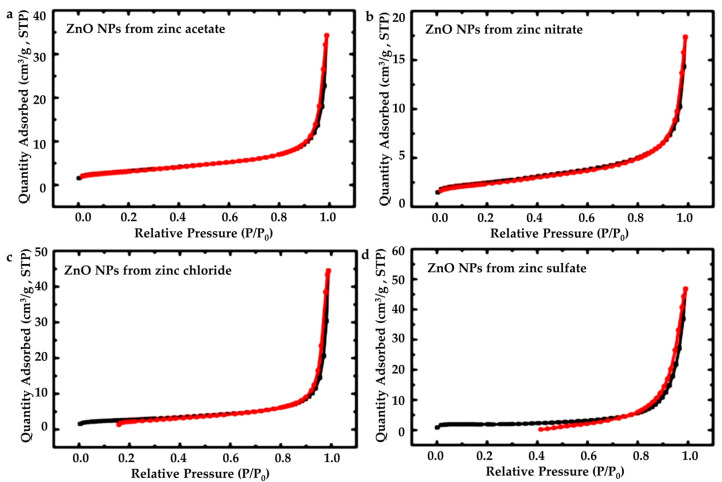
N2-isotherms for the ZnO nanoparticles, structured from all four precursors: (**a**) zinc acetate dehydrate (ZnAc), (**b**) zinc nitrate hexahydrate (ZnNit), (**c**) zinc chloride (ZnChlor), and (**d**) zinc sulphate heptahydrate (ZnSulf).

**Figure 6 nanomaterials-13-00122-f006:**
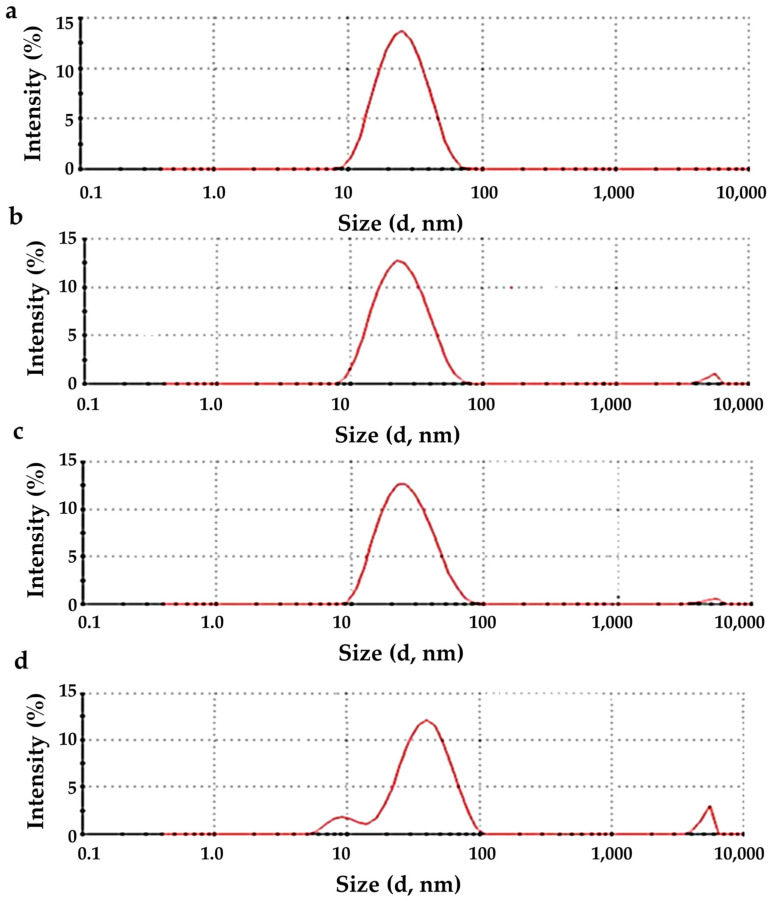
Size distribution diagram for the sample prepared from: (**a**) zinc acetate, (**b**) zinc nitrate, (**c**) zinc chloride, (**d**) zinc sulfate.

**Figure 7 nanomaterials-13-00122-f007:**
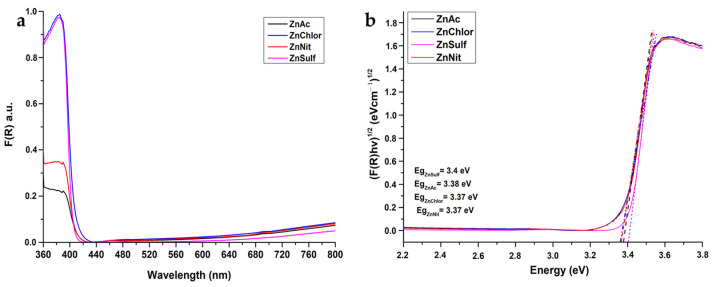
(**a**) F(R) reflectance as a function of the wavelength for the ZnO nanoparticles, structured from all four precursors (zinc acetate dehydrate (ZnAc), zinc nitrate hexahydrate (ZnNit), zinc chloride (ZnChlor), zinc sulphate heptahydrate (ZnSulf)). (**b**) Energy band gap of the same samples.

**Figure 8 nanomaterials-13-00122-f008:**
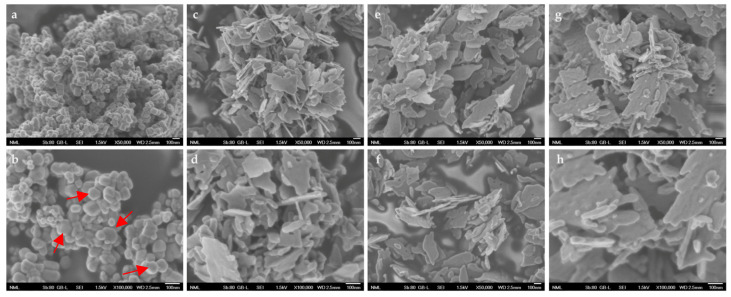
Representative FESEM images of produced ΖnO type particles prepared from: (**a**) zinc acetate (×50,000), (**b**) zinc acetate (×100,000), (**c**) zinc chloride (×50,000), (**d**) zinc chloride (×100,000), (**e**) zinc nitrate (×50,000), (**f**) zinc nitrate (×100,000), (**g**) zinc sulfate (×50,000), and (**h**) zinc sulfate (×100,000). Red arrows are used in order to point out some of the hexagonal-shaped particles in the case of ZnAc.

**Figure 9 nanomaterials-13-00122-f009:**
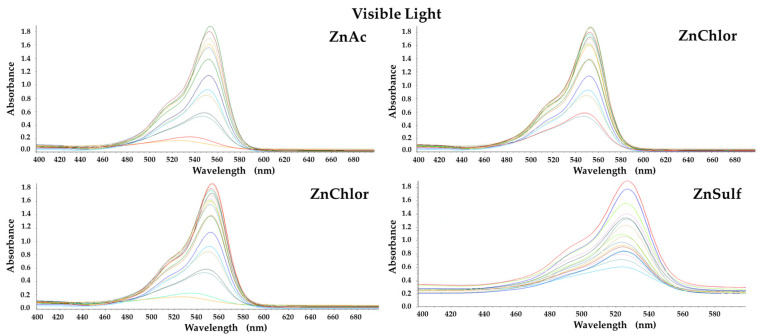
Real time UV–Visible spectra under visible light photocatalytic degradation of Rhodamine B, by the four ZnO powders, produced by different precursors.

**Figure 10 nanomaterials-13-00122-f010:**
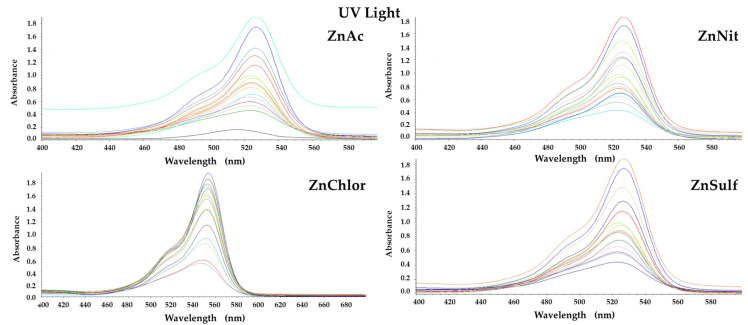
Real time UV–Visible spectra under UV photocatalytic degradation of Rhodamine B, by the four ZnO powders, produced by different precursors.

**Figure 11 nanomaterials-13-00122-f011:**
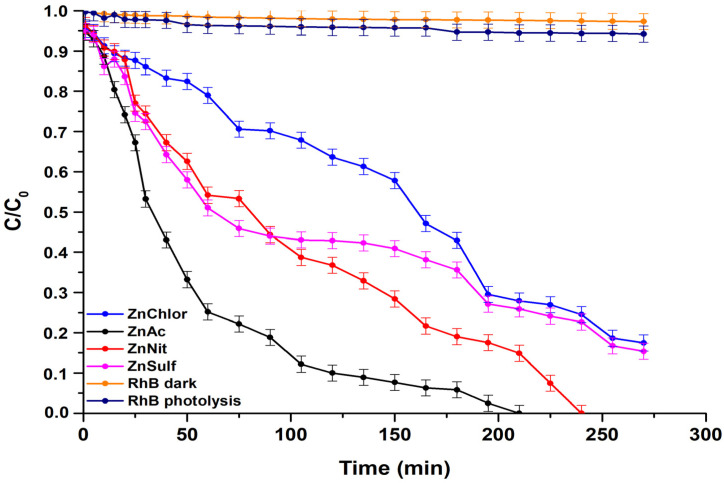
Degradation of Rhodamine B under visible light irradiation, by the four different ZnO powders, produced by different precursors.

**Figure 12 nanomaterials-13-00122-f012:**
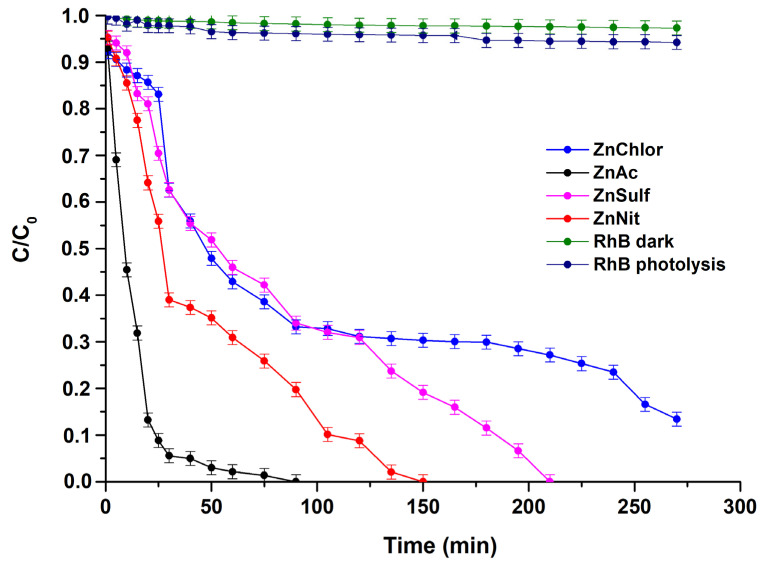
Degradation of Rhodamine B under UV irradiation, by the four different ZnO powders, produced by different precursors.

**Figure 13 nanomaterials-13-00122-f013:**
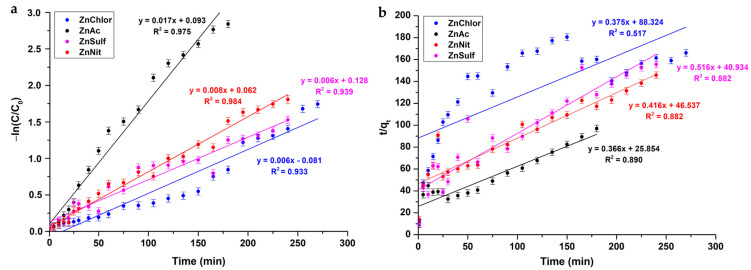
Photocatalytic kinetic model studies for the four different ZnO powders, produced by different precursors, following (**a**) a pseudo-first order model and (**b**) a pseudo-second order model, upon visible light photocatalysis.

**Figure 14 nanomaterials-13-00122-f014:**
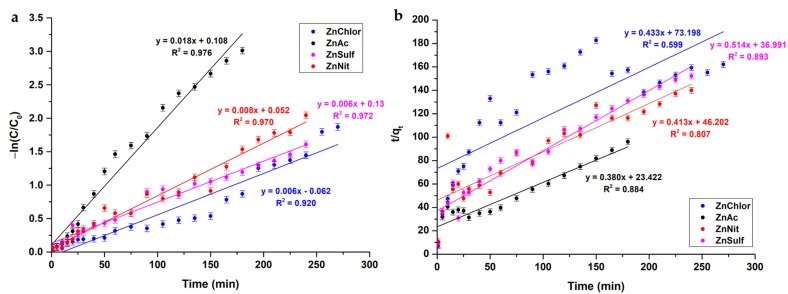
Photocatalytic kinetic model studies for the four different ZnO powders, produced by different precursors, following (**a**) a pseudo-first order model and (**b**) a pseudo-second order model, upon UV light photocatalysis.

**Figure 15 nanomaterials-13-00122-f015:**
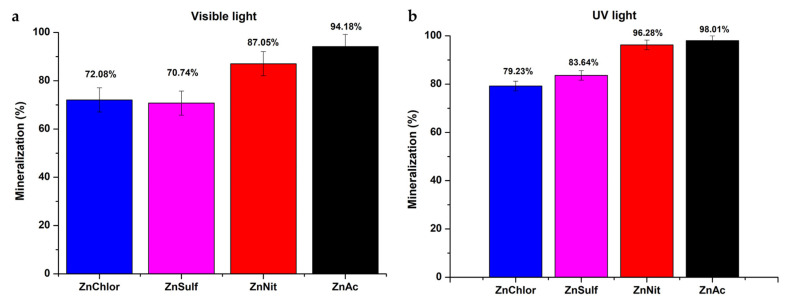
Mineralization (%) of each ZnO sample, obtained through TOC analysis, after the photocatalytic procedure under (**a**) visible light and (**b**) UV light.

**Figure 16 nanomaterials-13-00122-f016:**
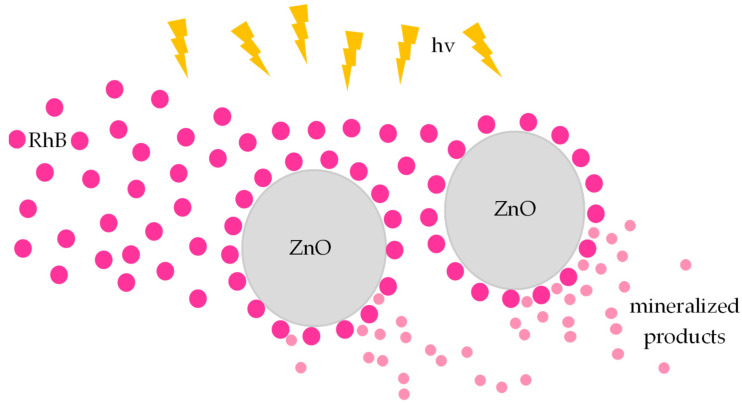
The main steps of the heterogeneous photocatalytic degradation of RhB by ZnO.

**Figure 17 nanomaterials-13-00122-f017:**
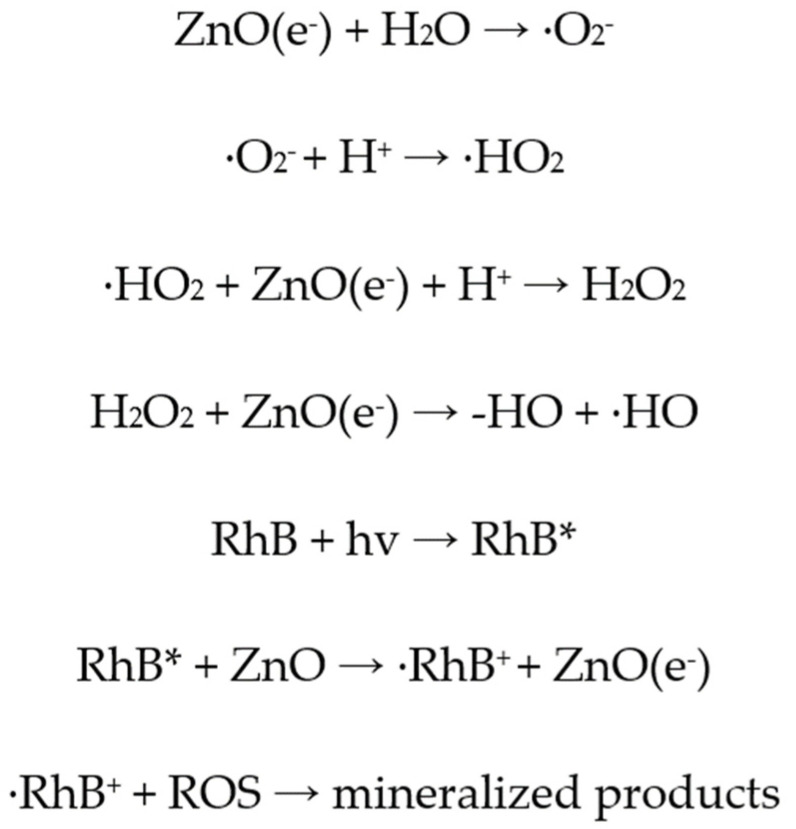
Photocatalytic degradation of RhB (chemical reactions).

**Figure 18 nanomaterials-13-00122-f018:**
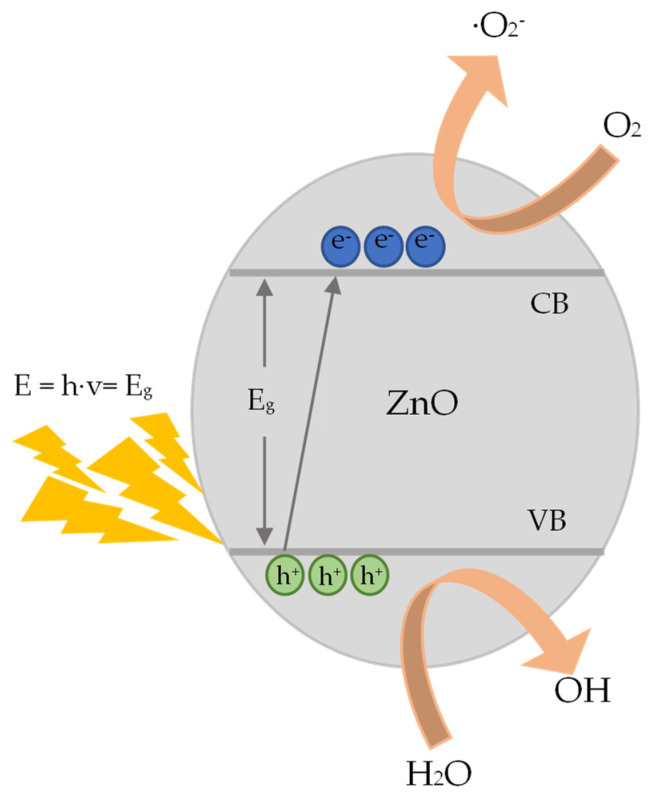
Representation of the photocatalytic process on the surface of ZnO.

**Figure 19 nanomaterials-13-00122-f019:**
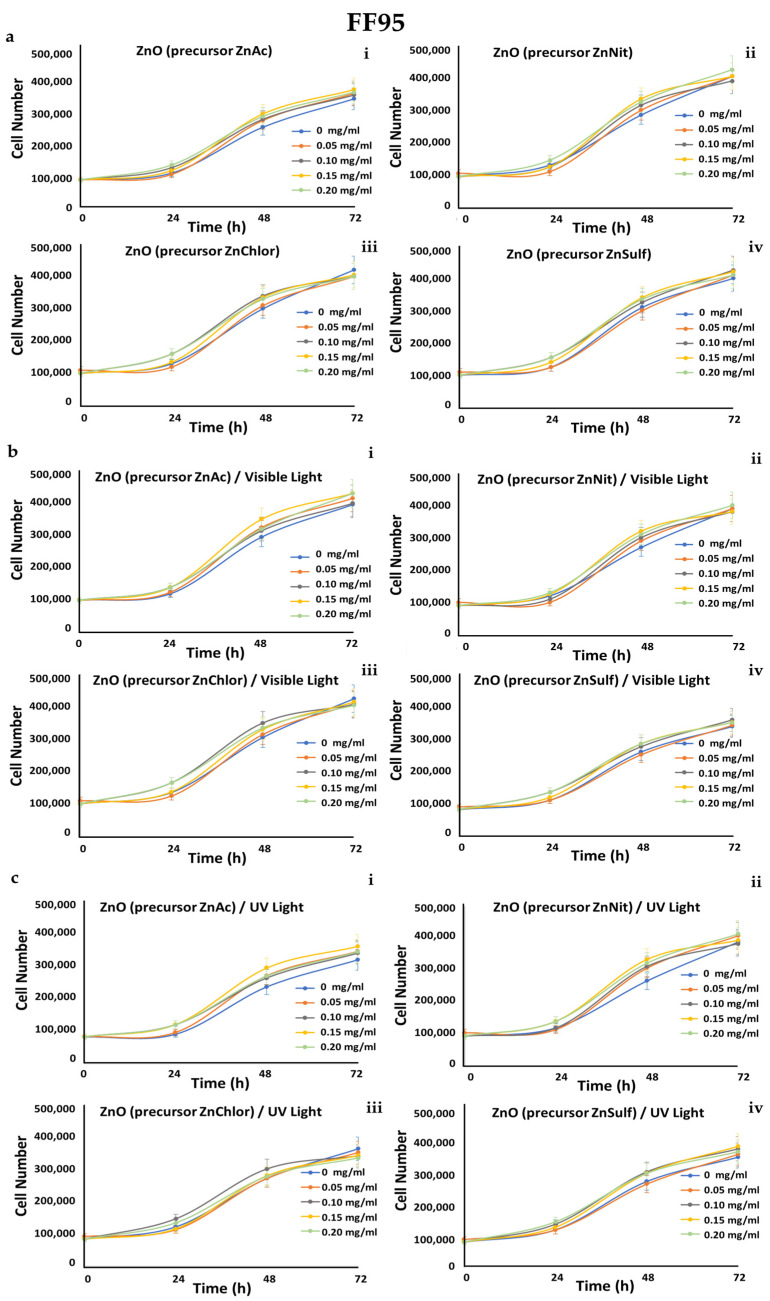
Growth rates of FF95 cell line in the presence of (**a**) zinc acetate dehydrate (ZnAc), zinc nitrate hexahydrate (ZnNit), zinc chloride (ZnChlor), zinc sulphate heptahydrate (ZnSulf). (**b**) Photo-activated with visible light ZnAc, ZnNit, ZnChlor, and ZnSulf. (**c**) Photo-activated with UV light ZnAc, ZnNit, ZnChlor, and ZnSulf. There is not any significant effect on cell proliferation. *p* < 0.05 was considered statistically significant.

**Figure 20 nanomaterials-13-00122-f020:**
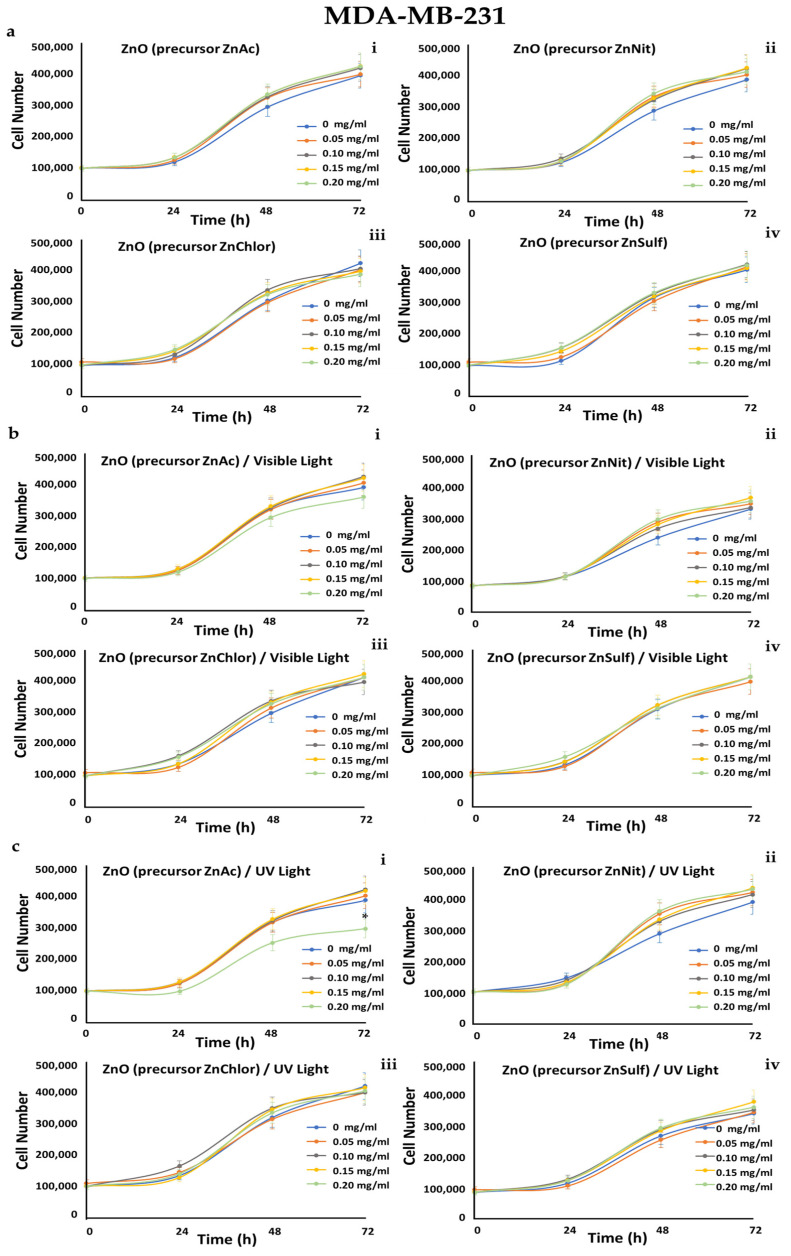
Growth rates of MDA-MB-231 cell line in the presence of (**a**) zinc acetate dehydrate (ZnAc), zinc nitrate hexahydrate (ZnNit), zinc chloride (ZnChlor), zinc sulphate heptahydrate (ZnSulf). (**b**) Photo-activated with visible light ZnAc, ZnNit, ZnChlor, and ZnSulf. (**c**) Photo-activated with UV light ZnAc, ZnNit, ZnChlor, and ZnSulf. There is not any significant effect on cell proliferation except from the photo-activated with UV light ZnAc that decreased the cell population by 25% at the concentration of 0.2 mg/mL. *p* < 0.05 was considered statistically significant.

**Figure 21 nanomaterials-13-00122-f021:**
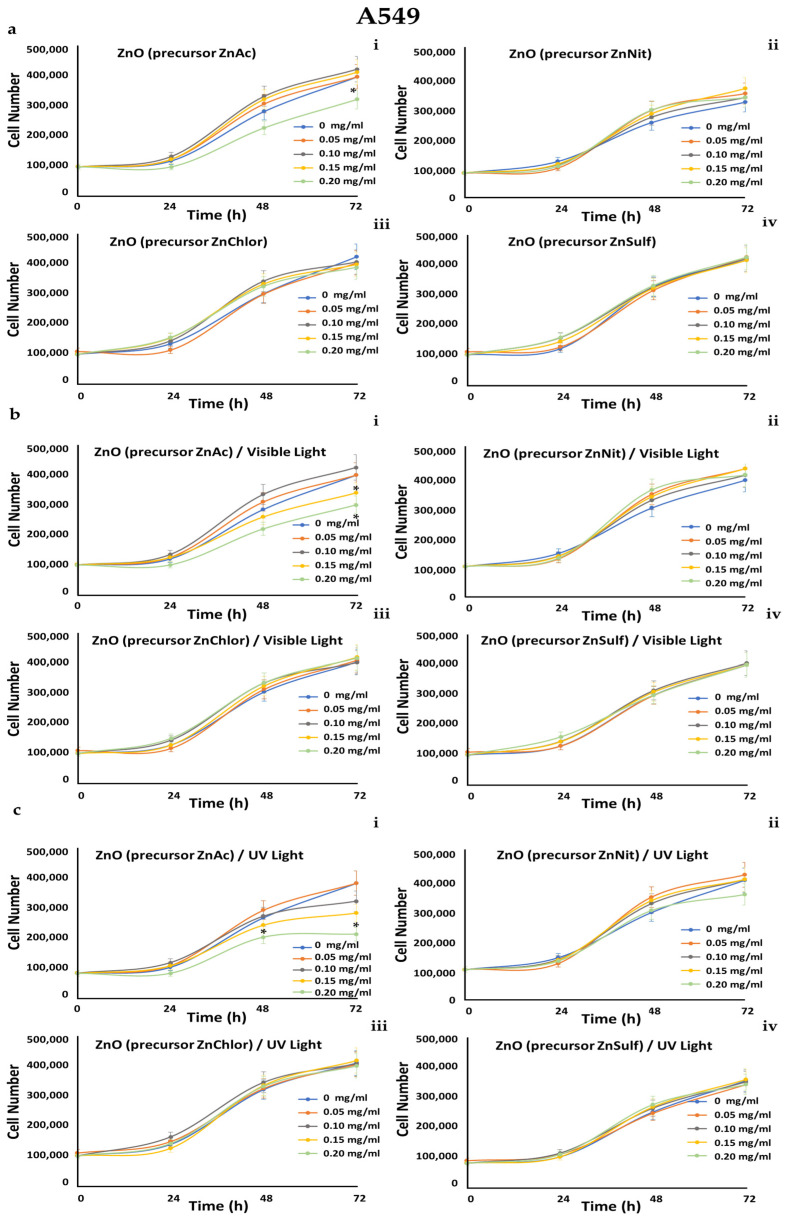
Growth rates of A549 cell line in the presence of (**a**) zinc acetate dehydrate (ZnAc), zinc nitrate hexahydrate (ZnNit), zinc chloride (ZnChlor), zinc sulphate heptahydrate (ZnSulf). (**b**) Photo-activated with visible light ZnAc, ZnNit, ZnChlor, and ZnSulf. (**c**) Photo-activated with UV light ZnAc, ZnNit, ZnChlor, and ZnSulf. ZnNit slightly decreased the cell population after photo-activation with UV, but this effect is not statistically significant. ZnAc has an important effect, reducing the cell number by 100,000 cells, at the concentration of 0.2 mg/mL. Photo-activation with visible light led to a further decrease of 0.15 and 0.2 mg/mL and even earlier (at 48 h) the UV-photoactivation allowed a significant decrease at the same concentrations. *p* < 0.05 was considered statistically significant.

**Figure 22 nanomaterials-13-00122-f022:**
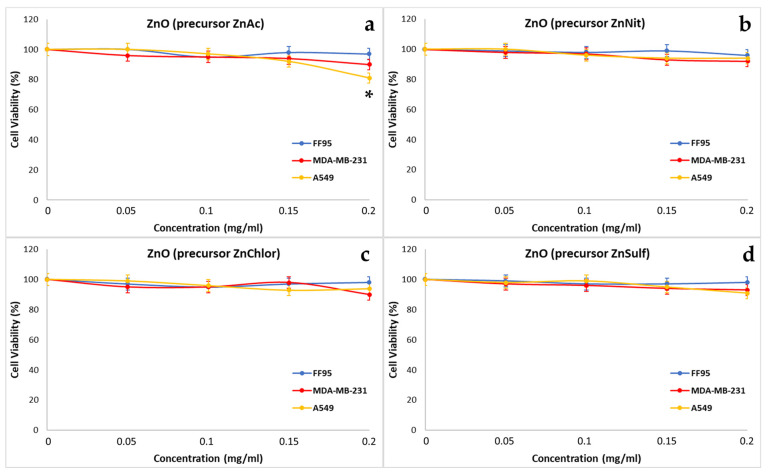
Effect of ZnO nanoparticles, synthesized using as precursor: (**a**) zinc acetate dehydrate (ZnAc), (**b**) zinc nitrate hexahydrate (in purple) (ZnNit), (**c**) zinc chloride (ZnChlor), and (**d**) zinc sulphate heptahydrate (ZnSulf) on FF95, MDA-MB-231, and A549 cells. There is not any significant decrease in cell population of any of the three cell lines because of the addition of ZnNit, ZnChlor, and ZnSulf, while ZnAc decreased the cell viability of A549 cells by ~20% at a concentration of 0.2 mg/mL. * *p* < 0.05 was considered as statistically significant.

**Figure 23 nanomaterials-13-00122-f023:**
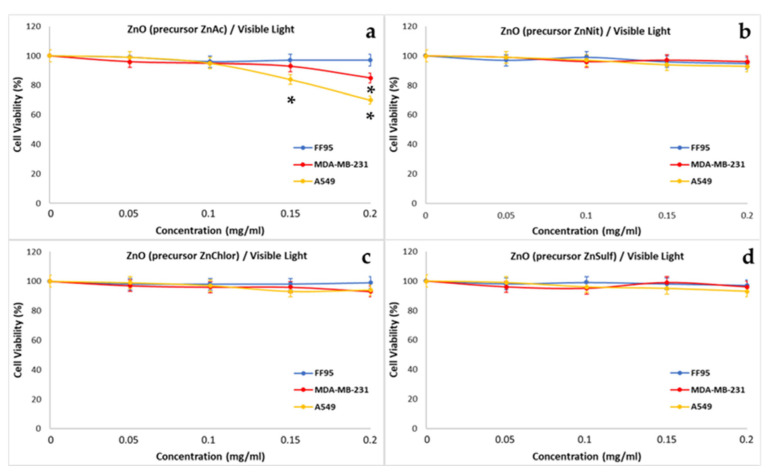
Effect of photo-activated with visible-light ZnO nanoparticles, synthesized using as precursor: (**a**) zinc acetate dehydrate (ZnAc), (**b**) zinc nitrate hexahydrate (ZnNit), (**c**) zinc chloride (ZnChlor), and (**d**) zinc sulphate heptahydrate (ZnSulf) on FF95, MDA-MB-231, and A549 cells. There is not any significant decrease in cell population of any of the three cell lines as a result of the addition of the photo-excitement of ZnNit, ZnChlor, and ZnSulf. Photo-activated ZnAc decreased the cell viability of A549 cells by ~30%, and 20% that of MDA-MB-23 at a concentration of 0.2 mg/mL. Furthermore, the cell viability of A549 was finally 85% after being treated with 0.15 mg/mL of ZnAc. * *p* < 0.05 was considered as statistically significant.

**Figure 24 nanomaterials-13-00122-f024:**
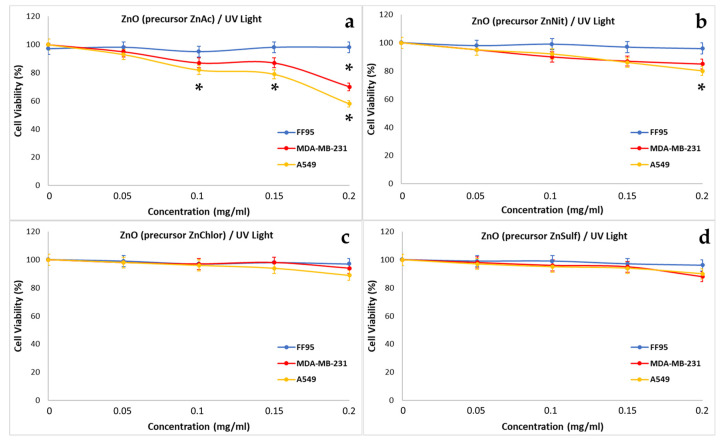
Effect of photo-activated with UV light ZnO nanoparticles, synthesized using as precursor: (**a**) zinc acetate dehydrate (ZnAc), (**b**) zinc nitrate hexahydrate (ZnNit), (**c**) zinc chloride (ZnChlor), and (**d**) zinc sulphate heptahydrate (ZnSulf) on FF95, MDA-MB-231, and A549 cells. There is not any significant decrease of cell population in any of the three cell lines as a result of the addition of the photo-excitement of ZnChlor and ZnSulf. A minor decrease in cell viability of A549 cells was observed with the addition of 0.2 mg/mL ZnNit. The same concentration of 0.2 mg/mL photo-activated ZnAc decreased the cell viability of A549 cells by ~42%, and MDA-MB-231 by 30%. Additionally, 0.1 and 0.15 mg/mL of photo-excited ZnNit could reduce the cell viability of A549 slightly. * *p* < 0.05 was considered as statistically significant.

**Table 1 nanomaterials-13-00122-t001:** Crystal lattice indices, average crystallite size, FWHM and crystallinity of the produced ZnO powders.

Sample ID	Crystal Lattice Index *(a = b ≠ c)*			Average Crystallite Size (nm) *	Full Width at Half Maximum (FWHM)	Crystallinity (%)
	*a*	*b*	*c*			
ZnAc	2.8137	2.8137	5.2087	4.270 ± 7.26·10^−3^	0.4215	81.99
ZnChlor	2.8149	2.8149	5.2105	5.470 ± 7.26·10^−3^	0.3291	76.05
ZnNit	2.8199	2.8199	5.2196	8.200 ± 7.26·10^−3^	0.2194	80.31
ZnSulf	2.8141	2.8141	5.2087	7.850 ± 7.26·10^−3^	0.2293	74.59

* Crystallite size was calculated according to the peak that was related to (101) plane.

**Table 2 nanomaterials-13-00122-t002:** d-spacing calculations for ZnAc powder.

Bragg’s Angle		d*_hkl_* (Å)	d*_hkl_* (nm)	*hkl*
2*θ*	*θ*			
31.78	15.89	2.8137	0.28137	100
34.44	17.22	2.6023	0.26023	002
36.26	18.13	2.4754	0.24754	101
47.57	23.78	1.9101	0.19101	102
56.63	28.31	1.6241	0.16241	110
62.90	31.45	1.4763	0.14763	103
66.54	33.27	1.4042	0.14042	200
68.00	34.00	1.3776	0.13776	112
69.12	34.56	1.3580	0.13580	201
72.64	36.32	1.3005	0.13005	202
76.99	38.49	1.2376	0.12376	004

**Table 3 nanomaterials-13-00122-t003:** d-spacing calculations for ZnChlor powder.

Bragg’s Angle		d*_hkl_* (Å)	d*_hkl_* (nm)	*hkl*
2*θ*	*θ*			
31.76	15.88	2.8149	0.28149	100
34.42	17.21	2.6032	0.26032	002
36.25	18.12	2.4762	0.24762	101
47.55	23.78	1.9106	0.19106	102
56.62	28.31	1.6243	0.16243	110
62.89	31.45	1.4766	0.14766	103
66.43	33.21	1.4063	0.14063	200
67.98	33.99	1.3778	0.13778	112
69.10	34.55	1.3583	0.13583	201
72.59	36.30	1.3012	0.13012	202
75.66	37.83	1.2559	0.12559	004

**Table 4 nanomaterials-13-00122-t004:** d-spacing calculations for ZnNit powder.

Bragg’s Angle		d*_hkl_* (Å)	d*_hkl_* (nm)	*hkl*
2*θ*	*θ*			
31.70	15.85	2.8199	0.28199	100
34.36	17.18	2.6078	0.26078	002
36.19	18.10	2.4799	0.24799	101
47.49	23.74	1.9131	0.19131	102
56.55	28.28	1.6260	0.16260	110
62.82	31.41	1.4781	0.14781	103
66.35	33.17	1.4077	0.14077	200
67.92	33.96	1.3790	0.13790	112
69.05	34.53	1.3591	0.13591	201
69.05	34.53	1.3591	0.13591	202
76.86	38.43	1.2394	0.12394	004

**Table 5 nanomaterials-13-00122-t005:** d-spacing calculations for ZnSulf powder.

Bragg’s Angle		d*_hkl_* (Å)	d*_hkl_* (nm)	*hkl*
2*θ*	*θ*			
31.77	15.89	2.8141	0.28141	100
34.44	17.22	2.6023	0.26023	002
36.26	18.13	2.4753	0.24753	101
47.57	23.78	1.9101	0.19101	102
56.62	28.31	1.6242	0.16242	110
62.91	31.45	1.4762	0.14762	103
66.42	33.21	1.4065	0.14065	200
67.99	34.00	1.3777	0.13777	112
69.12	34.56	1.3579	0.13579	201
71.76	35.88	1.3144	0.13144	202
76.00	38.00	1.2512	0.12512	004

**Table 6 nanomaterials-13-00122-t006:** Results of the BET method. (a) Micropore surface area via t-plot analysis, according to the Harkins and Jura model. (b) Cumulative volume of pores between 1.7 and 300 nm from N2-sorption data and the BJH desorption method. (c) Average pore diameter, calculated by the 4 V/σ method; V was set equal to the maximum volume of N2 adsorbed along the isotherm as P/Po → 1.0.

Sample ID	BET Surface Area (m^2^ g^–1^)	Micropore Surfuce Area (m^2^ g^–1^)	Cumulative Volume (1.7–300 nm) (cm^3^ g^–1^)	Average Pore Diameter (nm)
ZnAc	11	1	0.05	19
ZnChlor	9	2	0.07	29
ZnNit	8	2	0.02	13
ZnSulf	6	5	0.07	50

**Table 7 nanomaterials-13-00122-t007:** Results from DLS.

Sample ID	Hydrodynamic Diameter (D_h_)	PdI
ZnAc	25.9 ± 10.5	0.219
ZnChlor	41.3 ± 16.3	0.293
ZnNit	27.43 ± 1.19	0.177
ZnSulf	27.0 ± 10.7	0.157

**Table 8 nanomaterials-13-00122-t008:** Kinetic parameters of the ZnO powders under visible light photocatalysis.

Sample ID	Pseudo-First Order Kinetic Model		Pseudo-Second Order Kinetic Model	
	K_1_ (min^−1^)	R^2^	K_2_ (g·mg^−1^·min^−1^)	R^2^
ZnAc	0.017	0.975	0.366	0.890
ZnChlor	0.006	0.933	0.375	0.517
ZnNit	0.008	0.984	0.416	0.882
ZnSulf	0.006	0.939	0.516	0.882

**Table 9 nanomaterials-13-00122-t009:** Kinetic parameters of the ZnO powders under UV light photocatalysis.

Sample ID	Pseudo-First Order Kinetic Model		Pseudo-Second Order Kinetic Model	
	K_1_ (min^−1^)	R^2^	K_2_ (g·mg^−1^·min^−1^)	R^2^
ZnAc	0.018	0.976	0.380	0.884
ZnChlor	0.006	0.920	0.433	0.599
ZnNit	0.008	0.970	0.413	0.807
ZnSulf	0.006	0.972	0.514	0.893

## Data Availability

Not applicable.

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
