# Peer review of "ZnO Nanoparticles from Different Precursors and Their Photocatalytic Potential for Biomedical Use"

_nanomaterials, 2022, doi:10.3390/nano13010122_

Round 1

Reviewer 1 Report

This manuscript presents the “ZnO nanoparticles from different precursors and their photo-catalytic potential for biomedical use”. This manuscript lacks the innovations and details investigation of ZnO synthesis process as well as its application in photo-catalytic potential for biomedical use. Following comments should be addressed before publication:

1.       This manuscript contains so many grammatical errors. Some sentences are arranged improperly.

2.       The synthesis mechanism of ZnO should be explained in detail. What is the reason of high performance of ZnO Ac compared to others.

3.       Following articles can be used as references to explain the rational use of metal oxide in the introduction section:

Chemical Physics Letters Volume 804, October 2022, 139884

4.       The XRD spectrum in figure 1 is not clear. All the XRD panel should be plotted in same plots and the changes in the peak positions should be studied in detail.

5.       The specific surface area of ZnAc is 11 m2g-1. Authors should explain the reason for the high specific are of ZnAc compared to other samples.

6.       Highresolution FESEM images is necessary to study the morphology of materials. It seems that the ZnAc has smaller granular structure. The morphological changes can be explained.

7.       The study of the photocatalysis effect and cytotoxic effect of ZnO is very poor.

8.       Update the revised manuscript with the real time Uv-Visible spectra spectroscopy to stufy photocatalytic effects. The kinetics of photocatalysis should be studied in detail with the type of kinetic model. Also, investigate the isotherm of the photocatalysts.

9.       The revised manuscript should be updated with the schematic representations showing the photocatalysis process by ZNO.

10.   What about the band gap for ZnO obtained from different precursors?

11.   The instrument details and materials used in this work should be presented in the revised manuscript with their purity percentage.

Author Response

1
Response to Reviewer 1 Comments
The authors would like to thank the reviewer for the valuable comments and for giving them the opportunity to revise it.
They, also, want to underline that the motivation of this research is to compare the efficiency of the ZnO powders, produced by four different precursors, through the comparative determination of their physicochemical properties, the investigation of the photocatalysis effect under UV and visible light in parallel with cytotoxicity estimation. Even though, there are several studies focusing on the comparison of the obtained ZnO powders regarding the various synthesis routes (like thermal treatment), and others that investigate the photocatalytic effect of ZnO powders by studying various pollutants, to our knowledge, the present study for the first time, attempts to clarify the effect of using different precursors during the synthetic procedure under constant initial corresponding concentrations and thermal treatment conditions on the photocatalytic performance of the produced materials under visible and UV-radiation , as well as cytotoxicity and cell proliferation.
Since the authors understood, through the general comment of the reviewer that this study lacks the innovation, they added a new part at the end of the introduction (page 3, line 136-148) that highlights the points of novelty. Also, they added a sentence at the end of the abstract clarifying the novelty (page 1, lines 25-28).
All the modifications have been highlighted in yellow in the revised version of the manuscript. The reference numbering has also been changed. Also, the authors kept the same color motif in all the figures to help readers better comprehend the concept of the text.
Point 1: This manuscript contains so many grammatical errors. Some sentences are arranged improperly.
Response 1: The authors considering this comment, tried to correct the detected grammatical errors in the manuscript.
Point 2: The synthesis mechanism of ZnO should be explained in detail. What is the reason of high performance of ZnO Ac compared to others.
Response 2: The authors would like to thank the reviewer for this valuable and constructive comment. In order to help the potential readers to understand the synthesis mechanism of ZnO in detail, the authors prepared and added an extra figure (Figure 1, page 4, lines 169-172), that schematically represents the main steps of the synthesis procedure. Authors have explained in several parts of the revised manuscript and particularly at the section 3.2.1 and 3.2.2 that the reason of high performance of ZnAc compared to others seems to be associated with the relatively smaller crystalline size, the higher surface area and the hexagonal morphology, since these are the main points of difference, compared to the other ZnO powders.
2
Point 3: Following articles can be used as references to explain the rational use of metal oxide in the introduction section: Chemical Physics Letters Volume 804, October 2022, 139884
Response 3: The authors would like to thank the reviewer for this suggestion. They added at the beginning of the third paragraph of the second page, information related to this reference, with the appropriate citation (lines 68-71). All the reference numbering has changed.
Point 4: The XRD spectrum in figure 1 is not clear. All the XRD panel should be plotted in same plots and the changes in the peak positions should be studied in detail.
Response 4: The authors would gratefully thank the reviewer for this remark. They prepared and added a new figure (Figure 2, page 8). They also added information about the methodology that they followed to estimate several parameters, such as, average crystallite size, d-spacing, lattice indices, and crystallinity and Tables 1-5 (pages 8-10), including the obtained data and also a new part in the manuscript with a related discussion of these results (page 10).
Point 5: The specific surface area of ZnAc is 11 m2g-1. Authors should explain the reason for the high specific are of ZnAc compared to other samples.
Response 5: The authors would like to thank the reviewer for the constructive suggestion. They have added their explanation (page 12, last two lines) about the high specific area of ZnAc, based on the literature, correlating this parameter, with the relatively small average crystallite size of ZnAc, compared to the other powders, as it was obtained through XRD analysis. As the specific surface area is increased, a larger contact area between the catalyst and dye molecules can be established, which causes the maximum adsorbent effect. Similarly, it allows more surface to afford more reactive sites to the activity by setting higher chemical efficiency to the incident photons. The same can be useful in establishing the large number of active sites for the effective interaction between catalyst and the dye molecules (pages 619-624).
Point 6: Highresolution FESEM images is necessary to study the morphology of materials. It seems that the ZnAc has smaller granular structure. The morphological changes can be explained.
Response 6: The authors have taken into account this attentive remark. They replaced the previous SEM images, with new ones obtained from Highresolution FESEM (Section 3.1.7, Figure 8) and discussed the differences in the morphology (page 15).
Point 7: The study of the photocatalysis effect and cytotoxic effect of ZnO is very poor.
Response 7: The authors would like to thank the reviewer for this comment. They have enriched both the study of the photocatalysis effect (see section 3.2). and the cytotoxic effect of ZnO (see section 3.3).
3
Point 8: Update the revised manuscript with the real time Uv-Visible spectra spectroscopy to study photocatalytic effects. The kinetics of photocatalysis should be studied in detail with the type of kinetic model. Also, investigate the isotherm of the photocatalysts.
Response 8: The authors are thankful for this suggestion. They have added real time Uv-Visible spectra, thorough study of the kinetic model in detail and also TOC analysis for the estimation of RhB mineralization (3.2.1. and 3.2.2 Photocatalytic activity and Kinetics, pages 16-19 and 3.2.3. Mineralization of RhB, pages 19-20). Regarding the last remark of the reviewer, about the study of the isotherms, the authors would like to highlight that Figures 9 and 12 include also data obtained from the photolysis of RhB as well as its degradation under no illumination and there is no significant decrease in the concentration of RhB in absence of the catalyst giving evidence that the measured decrease in absorbance for the tested powders is due to the pure photodegradation process and not to sorption-desorption phenomena occurring on the surface of the ZnO catalyst. Additionally, the amount of Rhodamine B degraded during the photocatalytic procedure in the absence of each examined photocatalyst is extremely low. Therefore, the authors consider that the absorption isotherms are not applicable in this type of photocatalysis measurements based upon the experimental protocol that has been followed.
Point 9: The revised manuscript should be updated with the schematic representations showing the photocatalysis process by ZNO.
Response 9: The authors would like to thank the reviewer for this comment. They have added a new section (3.2.4. Photocatalysis Mechanism, page 20-21), with schematic representation of the photocatalysis process by ZnO and the related ROS reactions).
Point 10: What about the band gap for ZnO obtained from different precursors?
Response 10: The authors would appreciate this question and for this reason they have added a new section in the revised manuscript (3.1.6. Diffuse Reflectance UV-Vis Spectroscopy-page 14-15), with the energy band gap evaluation for all the ZnO powders.
Point 11: The instrument details and materials used in this work should be presented in the revised manuscript with their purity percentage.
Response 11: The authors would like to thank the reviewer for the kind comment and added instrument details and the purity percentage for the materials they used (2.1. Materials and Reagents – page 4, Instruments details in yellow on page 6).

Reviewer 2 Report

The article ZnO nanoparticles from different precursors and their photocatalytic potential for biomedical use shows the results of the characterization of the properties and evaluation of the practical application of ZnO nanoparticles obtained by the sol-gel method using various precursors. For characterization, the authors used a large number of different research methods, including X-ray diffraction, IR spectroscopy, scanning electron microscopy. To evaluate the photocatalytic properties, the classical organic dye Rhodamine B was used. In general, the proposed method for the synthesis of nanoparticles based on zinc oxide is not very new, since the use of this synthesis method has proven itself quite well as the main method of obtaining. The variations of reagents used for the synthesis are also quite well known, and the results of applicability in photocatalysis correlate quite well with the previously obtained results. According to the reviewer, this article may be of interest to a narrow circle of readers, specialists in this field, and despite the low level of scientific novelty, this work can be accepted for publication after the authors answer a number of questions that will allow, if they are fully answered. improve the submitted article.

1. Abstract. In the abstract, the authors should reflect the main difference between the presented work and other previously published works related to the synthesis of nanoparticles based on zinc oxide. It should also reflect the most significant and important points of the relevance of this study in comparison with other works.

2. The results of X-ray diffraction presented in the work require significant revision in presentation, in particular, the authors should significantly improve the quality of the presented diffraction patterns, as well as provide comparative data on changes in the position of the main diffraction reflections depending on the type of precursors. The structural parameters and the degree of crystallinity for the samples under study should be given.

3. There is not enough information about the declared structural changes and defects. This should be eliminated and the description of observed changes should be expanded. Comparative data on the parameters of the crystal lattice and the average size of crystallites are also not given.

4. Morphological studies of the sizes of the synthesized nanoparticles require their reduction to a single scale to determine the dynamics of change. It should also be noted that the presented nanoparticles have a spherical or close to spherical shape, while in most other works, zinc oxide nanoparticles are hexagonal or diamond-shaped structures. Authors should provide an explanation for such differences.

5. On the efficiency of photocatalytic decomposition, as a rule, Rhodamine B is quite stable in degradation and the efficiency of its decomposition does not exceed 80%, the authors should explain such a high photocatalytic ability of the synthesized nanoparticles, and also why this is connected. The results of mineralization should also be given.

6. The authors did not reflect the optical properties of the synthesized nanoparticles, which would make it possible to give an answer to such a high photocatalytic activity.

Author Response

1
Response to Reviewer 2 Comments
The authors would like to thank the reviewer for the valuable comments and for giving them the opportunity to revise it.
They, also, want to underline that the motivation of this research is to compare the efficiency of the ZnO powders, produced by four different precursors, through the comparative determination of their physicochemical properties, the investigation of the photocatalysis effect under UV and visible light in parallel with cytotoxicity estimation. Even though, there are several studies focusing on the comparison of the obtained ZnO powders regarding the various synthesis routes (like thermal treatment), and others that investigate the photocatalytic effect of ZnO powders by studying various pollutants, to our knowledge, the present study for the first time, attempts to clarify the effect of using different precursors during the synthetic procedure under constant initial corresponding concentrations and thermal treatment conditions on the photocatalytic performance of the produced materials under visible and UV-radiation, as well as cytotoxicity and cell proliferation.
Since the authors understood, through the general comment of the reviewer that this study lacks the innovation, they added a new part at the end of the introduction (page 3, line 136-148) that highlights the points of novelty. Also, they added a sentence at the end of the abstract clarifying the novelty.
All the modifications have been highlighted in yellow in the revised version of the manuscript. The reference numbering has also been changed. Also, the authors kept the same color motif in all the figures to help readers better comprehend the concept of the text.
Point 1: Abstract. In the abstract, the authors should reflect the main difference between the presented work and other previously published works related to the synthesis of nanoparticles based on zinc oxide. It should also reflect the most significant and important points of the relevance of this study in comparison with other works.
Response 1: The authors would like to thank the reviewer for this valuable and constructive comment. They have enriched the abstract by adding a sentence at the end, clarifying the novelty (page 1, lines 25-28). Also, they added a new part at the end of the introduction (page 3, line 136-148) that highlights the points of novelty.
Point 2: The results of X-ray diffraction presented in the work require significant revision in presentation, in particular the authors should significantly improve the quality of the presented diffraction patterns, as well as provide comparative data on changes in the position of the main diffraction reflections depending on the type of precursors. The structural parameters and the degree of crystallinity for the samples under study should be given.
Response 2: The authors would gratefully thank the reviewer for this remark. They prepared and added a new figure (Figure 2, page 8). They also added information about the methodology
2
that they followed to estimate several parameters, such as, average crystallite size, d-spacing, lattice indices, and crystallinity and Tables 1-5 (pages 8-10), including the obtained data and also a new part in the manuscript with a related discussion of these results (page 10).
Point 3: There is not enough information about the declared structural changes and defects. This should be eliminated, and the description of observed changes should be expanded. Comparative data on the parameters of the crystal lattice and the average size of crystallites are also not given.
Response 3: The authors would like to thank the reviewer for this suggestion. They have added new images obtained from High resolution FESEM (Section 3.1.7, Figure 8) and discussed the differences in the morphology (page 15). Furthermore, as it is already mentioned, they added information about the methodology that they followed to estimate several parameters, such as, average crystallite size, d-spacing, lattice indices, and crystallinity and Tables 1-5 (pages 8-10), including the obtained data and also a new part in the manuscript with a related discussion of these results (page 10).
Point 4: Morphological studies of the sizes of the synthesized nanoparticles require their reduction to a single scale to determine the dynamics of change. It should also be noted that the presented nanoparticles have a spherical or close to spherical shape, while in most other works, zinc oxide nanoparticles are hexagonal or diamond-shaped structures. Authors should provide an explanation for such differences.
Response 4: The authors would like to thank the reviewer for this valuable suggestion. They have added new images obtained from High-resolution FESEM (Section 3.1.7, Figure 8) and discussed the differences in the morphology (page 15). The shape was previously characterized as spherical, since high-resolution FESEM and deeper analysis unraveled that ZnAc exhibited a combination of almost spherical and hexagonal particles, while all the other ZnO powders were flake-shaped. It is widely used that ZnO is found in various morphologies, including 0D, 1D, 2D and 3D structures and various parameters can affect the observed morphology. The authors have attempted to correlate these differences with the selection of the precursor during the synthesis procedure, since all the other experimental conditions remained constant. Also, it is worth mentioning that a posteriori and through photocatalysis and cytotoxicity test hexagonal shaped have proven more effective.
Point 5: On the efficiency of photocatalytic decomposition, as a rule, Rhodamine B is quite stable in degradation and the efficiency of its decomposition does not exceed 80%, the authors should explain such a high photocatalytic ability of the synthesized nanoparticles, and also why this is connected. The results of mineralization should also be given.
Response 5: The authors are thankful for the comment. They have enriched the revised manuscript with real time Uv-Visible spectra, thorough study of the kinetic model in detail accompanied by TOC analysis for the estimation of RhB mineralization (3.2.1. and 3.2.2, pages 16-19 and 3.2.3. Mineralization of RhB).
The authors agree with reviewer about the stability of Rhodamine B, since previous studies of the corresponding author have focused on this issue (Koukouzelis et al. 2020, Ionic liquid –
3
Assisted synthesis of silver mesoparticles as efficient surface enhanced Raman scattering substrates, Journal of Molecular Liquids, 306, 112929, https://doi.org/10.1016/j.molliq.2020.112929). However, upon photocatalysis experiments stability of Rhodamine is not a rule, since several studies have shown that the photocatalytic degradation of Rhodamine B can be achieved during the first 3-4 hours. The results of this study are in accordance with various others (e.g., Yang et al. 2018, Photocatalytic degradation of rhodamine B catalyzed by TiO2 films on a capillary column, RSC Advances, 8, 11921-11929, https://doi.org/10.1039/C8RA00471D, Nagaraja et al. 2012, Photocatalytic degradation of Rhodamine B dye under UV/solar light using ZnO nanopowder synthesized by solution combustion route, Powder Technology, 215–216, 91-97, https://doi.org/10.1016/j.powtec.2011.09.014, Sundararajan et al. 2017, Photocatalytic degradation of rhodamine B under visible light using nanostructured zinc doped cobalt ferrite: Kinetics and mechanism, Ceramics International, 43(1), 540-548, https://doi.org/10.1016/j.ceramint.2016.09.191, etc.).
Point 6: The authors did not reflect the optical properties of the synthesized nanoparticles, which would make it possible to give an answer to such a high photocatalytic activity.
Response 6: The authors appreciate reviewers’ suggestion and they have added a new section in the revised manuscript (3.1.6. Diffuse Reflectance UV-Vis Spectroscopy-page 14-15) devoted to the energy band gap evaluation for all the ZnO powders. Since finally Eg was not significantly different for the four ZnO powders, they have attributed the observed high catalytic behavior of ZnAc due to reduced average crystallite size, higher surface area and the hexagonal morphology of the ZnAc, compared to the other ZnO powders.

Round 2

Reviewer 1 Report

The revised manuscript can be accept in present form.

Reviewer 2 Report

The authors answered all the questions, the article can be accepted for publication.